



# The Met Office operational wave forecasting system: the evolution of the Regional and Global models

Nieves G. Valiente[1], Andrew Saulter[1], Breogan Gomez[1], Christopher Bunney[1], Jian-Guo Li[1], Christine Pequignet[1], Tamzin Palmer[1]

[1]Met Office, Fitzroy Road, EX1 3PB, Exeter, UK

*Correspondence to*: Nieves G. Valiente (nieves.valiente@metoffice.gov.uk)

**Abstract.** The Met Office operational wave modelling system is an operational forecast system run four times daily at the Met Office to provide global and regional forecasts up to 7 days ahead. The underpinning model uses a recent development branch of the spectral wave model WAVEWATCH III® (version 7.12) that includes a number of updates developed at the Met Office.
Code contributions include the Spherical Multiple-Cell (SMC) grid, rotated pole grid formulation for mid latitudes, enhancements to OASIS coupling and updates to the netCDF postprocessing. Here we document and describe the technical details behind the Met Office operational system of WAVEWATCH III® configurations with a view to further development. These include a global forecast deterministic model (GS512L4EUK) and two regional models nested one-way covering the Northwest (NW) European shelf and UK waters (AMM15SL2) as well as an Atlantic wave ensemble (AS512L4EUK).
GS512L4EUK and AS512L4EUK are based on a four-tier SMC 25-12-6-3km grid refinement where currents are not included. AMM15SL2 is run operationally both as a standalone forced model and as the wave component of a two-way ocean-wave coupled operational system FOAM-AMM15. The AMM15SL2 baseline configuration is based on a two-tier SMC grid that focuses on the shelf seas around the United Kingdom (3km resolution) where coastal cells have 1.5km resolution and wave-current interaction is included. Results from a 2-year hindcast demonstrate the ability of the baseline configurations to
reproduce both in-situ and satellite wave observations. Model-observations correlation is above 0.94–0.96 with standard deviations of differences that correspond to maximum 13–25% of the observed mean bulk wave diagnostics, demonstrating the quality and accuracy of the system. Evidence of resolution dependent differences in wave growth was observed, leading to slightly overestimated significant wave heights when replicating coastal mid-range conditions by AMM15SL2, and better suited to replicate the extremes. Additionally, the inclusion of wave-current interaction in AMM15SL2 tends to larger spread
on the observation-model differences. Hence, although a positive impact of the surface currents is not always shown in the overall statistics of the significant wave height, the addition of currents helps to significantly improve the prediction of the wave direction and period near the coast (>20% improvement), which has implications in beach safety, risk to coastal overtopping and shoreline evolution. Future system developments such as the use of sea point wind forcing, the optimisation of the models in line with model resolution, the utilisation of SMC multigrid and data assimilation are discussed.





## 1 Introduction

Marine monitoring and prediction are crucial for the coastal and offshore sectors. Having an accurate short range wave forecast is essential in many different marine and coastal applications. A wide range of areas such as marine navigation or offshore industries (e.g., renewable energy offshore farms, fishing, commercial oil and gas extraction) rely on accurate forecasts to ensure a safe and timely functioning of their activities. Forecasting of dangerous events that may lead to human and property

risk both offshore and at the coast is key for rapid decision making. Numerical weather prediction (NWP) models are used for operational weather and ocean forecasting, providing model outputs to downstream users and forecasters. Met Office NWP systems for ocean forecasting include forecasts of ocean dynamics, waves, storm surges and ecosystems. Hence, Met Office operational ocean forecasting models deliver prediction and monitoring of the marine environment contributing to safety at sea, industry and marine planning among others (Siddorn et al., 2016).

The National Centres for Environmental Prediction (NCEP) community spectral model WAVEWATCH III® (Tolman, 2014), herein WW3, is used operationally in both global and regional model configurations worldwide (e.g., GFSv16 wave (NOAA, 2020)). The Met Office have developed an operational system of WW3 configurations that is based on a recent development branch of the community code (version 7.12). As part of the WW3 Development Group (WW3DG), the Met Office has contributed with several developments to the WW3 codebase, including:

• the Spherical Multiple-Cell (SMC) grid which provides an unstructured multi-resolution (i.e., coarser offshore with higher resolution in coastal waters) spatial grid (Li, 2012) to improve model efficiency and enable improved forecast skill toward coastal zones;

• rotated pole grid formulation for mid latitudes;

• enhancements to the OASIS coupling for compatibility with ocean and atmospheric models;

• and updates to the netCDF postprocessing. These include grid interpolation from SMC to regular grids for products generation, CF compliant netCDF and user configurable netCDF meta data to maintain consistency with Copernicus Marine service naming conventions.

These latest developments have facilitated the migration to version 7.12 of WW3 in wave forecasting operations and in research coupled models including Northwest European shelf (e.g., Lewis et al., 2019; Bruciaferri et al., 2021) and the Indian regional

(Castillo et al., 2022) coupled wave-ocean and atmosphere-wave-ocean research systems. This paper documents the latest WW3 wave model developments introduced by the Met Office and describes the Met Office WW3 based wave operational forecasting system which includes a global model and two regional models nested one-way covering the Northwest European shelf and United Kingdom (UK) waters and the Atlantic wave ensemble. A detailed description of the operational wave modelling system is presented in Sect. 2. Model evaluation performance with a view to further development is described in

Sect. 3. Model analysis is focused on the global and regional UK waters baseline configurations. Operational forecast skill is shown in Sect. 4. Finally, a summary with key challenges and future work to update the systems is presented in Sect. 5.





## 2 The Met Office operational wave models

The Met Office operational forecasting system of WW3 configurations includes a global forecast deterministic model (GS512L4EUK) and regional deterministic models nested one-way covering the Northwest (NW) European shelf and UK

waters (AMM15SL2) as well as an Atlantic wave ensemble (AS512L4EUK) (Fig. 1). AMM15SL2 is run as waves standalone (i.e. forced one-way by winds and surface currents) and as the wave component of the ocean-wave coupled operational system FOAM-AMM15 (e.g., Bruciaferri et al., 2021; Lewis et al., 2019) used to produce Copernicus Marine Service products (Saulter, 2020b) from the Northwest Shelf Monitoring and Forecast Centre (NWS-MFC; e.g.: https://marine.copernicus.eu/about/producers/nws-mfc).

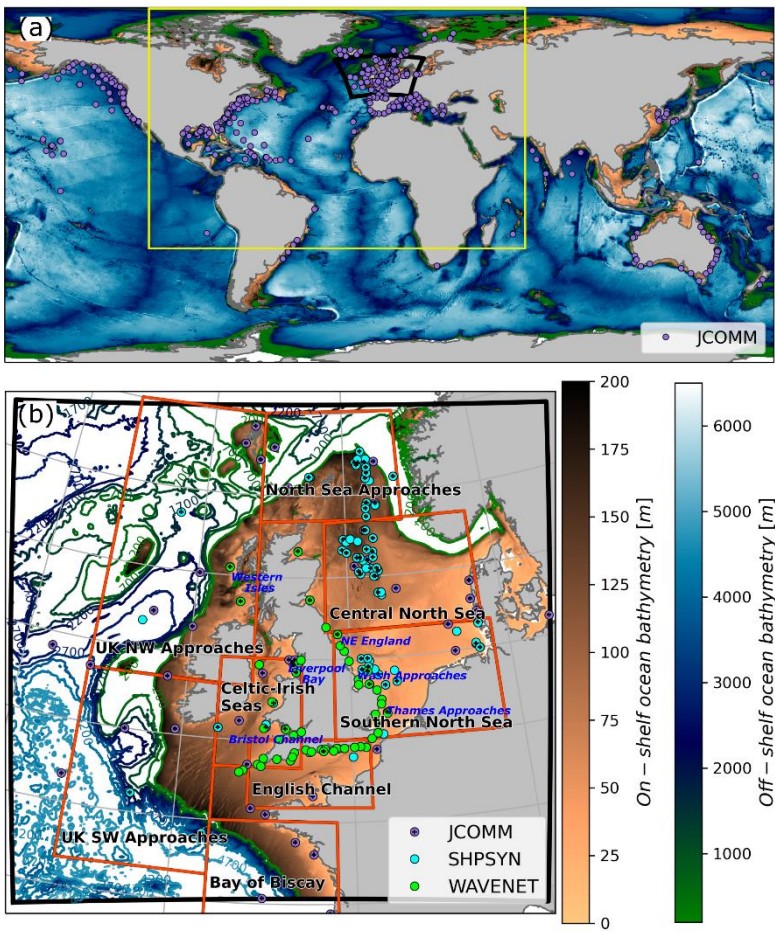

**Figure 1. (a) Global and (b) northwest (NW) European shelf – UK waters physical context and model domains. Yellow box and black solid line in (a) indicate the cut-off area for the Atlantic wave ensemble and the NW European shelf – UK waters domains, respectively. (b) NW European shelf – UK waters domain with areas used for analysis indicated in red. In-situ observations are shown as solid dots. In-situ observations include the Joint WMO IOC Commission for Oceanography and Marine Meteorology's**
**operational Wave Forecast Verification Scheme (JCOMM), Ship Synop Ob-servations at fixed platforms (SHPSYN) and UK WAVENET and National Network of Regional Coastal Monitoring Programmes in-situ observations for coastal waters (WAVENET). Locations where there is overlap with JCOMM observations are marked with a cross.**



## 2.1 Research to operations

All mission critical NWP models at the Met Office are run under an operationally maintained supercomputer production system known as the Operational Suite (OS). To maintain consistency and operational resilience, scientific and technical updates to these models follow a prescribed process defined in Parallel Suite (PS) projects, which aim to ensure the successful pull through of scientific improvements of the Met Office's Numerical Weather Prediction Models into the Operational environment (Walters, 2021). For the upstream NWP modelling systems a PS is essentially a copy of the latest operational suite to which

scientific and technical updates are applied. The PS is run in parallel with the current Operational Suite on a separate HPC to avoid any resource contention.  Once the PS is stable it will be "frozen" and cycled for a 6-8 week period during which verification and performance metrics will be collected. Once all the system performance checks of the PS are concluded this becomes the OS. Both OS and PS are numbered sequentially. The models described here correspond to the latest Met Office operational systems that became operational in May 2022 after Parallel Suite 45 (herein PS45), run in parallel with Operational

Suite 44 (hereafter OS44). All these suites are built as a rose suite workflow - a toolkit for writing, editing and running application configurations (http://metomi.github.io/rose/doc/html/index.html, last access: 01 July 2022) - where the model components, configurations and running characteristics are defined. Refer to *Sect. 2.4* for more detail.

## 2.2 Core model description

The Met Office operational wave forecasting system is based on the WAVEWATCH III® spectral model (Tolman, 2014) at

version 7.12. This model resolves the evolution of the two-dimensional (frequency-direction) wave energy spectrum in time and space using the net effect of sources and sinks of wave energy, i.e. total source term. The total source term can be defined as the combination of different physical processes that in deep waters can be simplified to the wind-wave interaction term, a nonlinear wave-wave interaction term and a dissipation term (Valiente et al., 2021a). Additionally, the operational systems include a linear input term for initial wave growth and additional shallow water processes (i.e., depth-induced breaking and

wave bottom interactions). Refer to *Supplement Material* for a detailed information of the model source terms parameterisations and the compilation switches.

The Met Office operational wave forecasting systems use the Ardhuin et al. (2010) ST4 package to parameterise wave growth and dissipation via whitecapping. The family of parameterisations in ST4 uses a positive part of the wind input based on WAM cycle 4 parameterisation (Janssen, 2004) with an ad-hoc reduction of the wind contribution to account for the impact of long

waves on short waves through a tuneable sheltering coefficient (TAUWSHELTER=0.3; refer to Table S2 in Supplement material) that decreases the drag coefficient at high winds (Saulter et al., 2017; Valiente et al., 2021a). For compatibility with Met Office Global Unified Model wind forecast data, a minor adjustment on the control of the input wind stress (BETAMAX on namelist of 1.39; refer to Table S2 in Supplement material) has been implemented across both global and regional wave





models. The BETAMAX value is also adjusted for the particular case of the ocean-wave regional coupled configuration which
is forced by ECMWF winds (BETAMAX on namelist of 1.48; refer to Table S2 in *Supplement material*).

As part of the shallow water physics, the Met Office wave systems include source terms to resolve depth induced refraction, shoaling and breaking. Hence, shallow water wave energy dissipation includes the surf breaking parameterisation proposed by Battjes and Janssen (1978) (DB1) and JONSWAP bottom friction formulation (BT1; Hasselmann et al., 1973). The Discrete Interaction Approximation (DIA) package (NL1; Hasselmann et al., 1985) is used to resolve nonlinear wave–wave quadruplets
interactions. NL1 is developed for deep water, using the appropiate dispersion relation in the resonance conditions. For shallow water this source term uses a scaled version of the deep-water dispersion relation. Conversion from wind speed to momentum stress flux computations are included in the source term (FLX0); i.e., stress is defined implicitly inside the source terms subroutines. Additionally, a switch with linear wave growth (LN1; Cavaleri and Rizzoli, 1981) for lower winds is also implemented (Valiente et al., 2021b). LN1 allows for the consistent spin-up of the model from calm conditions and improves
the initial wave growth. Model spectral resolution is identical in all the wave operational systems with 30 frequencies logarithmically spaced between 25 to 1.5 seconds (starting at 0.04118Hz) and 36 directional bins that are linearly spaced.

All configurations of the Met Office operational forecasting system utilise the SMC grid (Li, 2012). One of the key features of this grid is that it allows higher resolution cells in areas of interest (shallow water, coastal areas and islands) whilst maintaining coarse resolution in the open ocean for computational efficiency. The SMC grid retains the quadrilateral cells as
in the standard latitude-longitude grid so that simple finite difference schemes could be used. Sub-time-steps are applied on different cell sizes to speed up propagation calculations with a choice of 2nd or 3rd order upstream non-oscillatory (UNO) advection schemes (Li, 2008). The refraction induced wave spectral rotation and the great circle turning are combined and calculated with a re-mapping scheme, which is not subject to the CFL restriction but to a physical limit not exceeding the bathymetry gradient direction or a user defined limit angle. Grid cells are merged at high latitudes to relax the CFL restriction
and a fixed reference direction is used to define wave spectra in the polar region so that the whole Arctic Ocean could be included in the global domain. The multi-resolution refinement is useful to resolve small islands and coastline details, which are important in ocean surface wave propagations (Saulter et al., 2017). 'Garden Sprinkler Effect' (GSE) caused by the discrete directional bins of the wave energy spectrum is alleviated with a diffusion term similar to the PR2 option in WW3 model (Booij and Holthuijsen, 1987), plus an optional averaging scheme for further smoothing (WW3DG, 2019).

**2.3 Configurations of the operational forecasting system**





**Table 1 Specifications of the operational production of the Met Office wave systems: GS512L4EUK, AS512L4EUK, AMM15SL2 and AMM15 coupled.**

| | | Forecast Run | Update Run |
|---|---|---|---|
| **GS512L4EUK** | *Forecast length and run frequency* | T+144 for 0ZZ, 12Z<br>T+66 for 6Z, 18ZZ | T+6 for 0Z, 6Z, 12Z, 18Z |
| | *Wind forcing* | Hourly NWP global forecast at 10km resolution | Hourly NWP global update at 10km resolution |
| | *Ice forcing* | Global OSTIA at 1/12º resolution | Global OSTIA at 1/12º resolution |
| | *Initialisation* | Restart file update T+6 | Restart file update T+6 |
| | *Boundary conditions* | -- | -- |
| **AS512L4EUK** | *Forecast length* | T+168 | -- |
| | *Run frequency and members* | 00Z/12Z:0-17 members, 06Z/18Z:0,18-34 members | -- |
| | *Wind forcing* | Hourly NWP MOGREPS-Global forecast atmospheric ensemble at 20km resolution | -- |
| | *Ice forcing* | Global OSTIA at 1/12º resolution | -- |
| | *Initialisation* | Restart file update T+6 | -- |
| | *Boundary conditions* | 2D spectral boundary conditions at 25 km resolution | -- |
| **AMM15SL2** | *Forecast length and run frequency* | T+66 for 0Z, 06Z, 12Z, 18Z | T+6 for 0Z, 6Z, 12Z, 18Z |
| | *Wind forcing* | Hourly NWP global forecast at 10km resolution | Hourly NWP global update at 10km resolution |
| | *Current forcing* | Hourly AMM15 (00Z) at 1.5km resolution | Hourly AMM15 (00Z) at 1.5km resolution |
| | *Initialisation* | Restart file update T+6 | Restart file update T+6 |
| | *Boundary conditions* | 2D spectral boundary conditions at 25 km resolution | 2D spectral boundary conditions at 25 km resolution |
| **AMM15 coupled** | *Forecast length* | T+144 | -- |
| | *Run frequency* | 00Z | -- |
| | *Wind forcing* | Hourly ECMWF forecast winds for T0 to T72<br>3-hourly ECMWF forecast winds for T72 to T144 | -- |
| | *Initialisation* | T-48 hindcast cycle. Restart file T-24 | -- |
| | *Boundary conditions* | 2D spectral boundary conditions at 25 km resolution | -- |
| | *Hindcast* | T-48 using hourly ECMWF winds from previous analysis cycle | -- |

### 2.3.1 GS512L4EUK Global Wave Model

The wave forecast global model configuration GS512L4EUK covers the whole globe from 80° S to 86° N (Fig. 1a) and model bathymetry is based on GEBCO 2014. The model grid is based on a four-tier SMC 25-12-6-3km grid refinement where the coarsest cells are located in open waters and resolved at approximately 25km (0.35° longitude by 0.23° latitude) in mid-latitudes. The 25-km coarsest cells are then halved to 12km, 6km and 3km as the grid gets closer to the coastline. The configuration is denoted as GS512L4EUK to represent a base resolution equivalent to N512 atmosphere, the use of the SMC

grid with four-tier levels, and the designation of special interest areas in UK waters where the higher resolution is applied (Saulter et al., 2016). This configuration was first introduced in November 2016 at OS38 (Saulter et al., 2016).





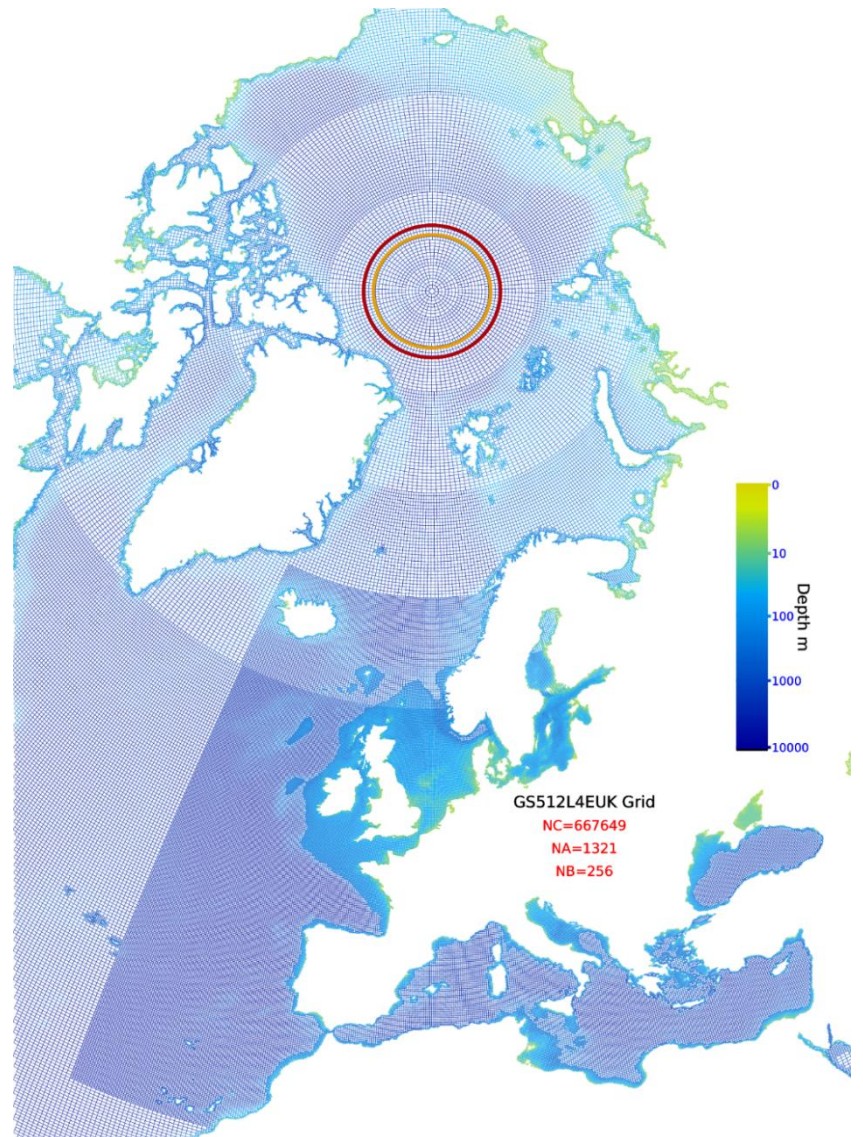

**Figure 2. Spherical Multiple-Cell (SMC) GS512L4EUK global model grid across the European-Arctic region. Coarsest (open waters) cells are resolved at approximately 25km in mid-latitudes (0.35° longitude by 0.23° latitude) and reduced by a factor of two to 12km, 6km and 3km. 12km cell size are set over the European region (27N 25W to 68N 42E) with a reduction to 6km for cells with depths less than 320m depth and to 3km for those cells around the UK coastline.**

Fig. 2 shows the European-Arctic region of the GS512L4EUK global model grid. The use of the 3km cell refinement has been restricted to waters of the NW European shelf (45° N 16° W to 61.15° N 9.4° E) in order to reduce computational costs. This cell resolution is only applied to coastal waters to best represent the coastal mask (Saulter et al., 2016). Over the wider European region (covering approximately 25° W to 27° E and 42° N to 68° N), the coarsest cell size has been set to 12km so that the model can exploit the full detail of the current Met Office global atmosphere-ocean model (approximately 10km resolution), whilst any cells with depths less than 320m are resolved at 6km. This depth was chosen as a threshold to apply higher resolution



since wave energy with mean periods of about 18s or longer will begin to interact with the seabed. At higher latitudes,
longitudinal cell sizes are doubled (by a factor of two at 60° N, four at 75° N, eight at 83° N) in order to support a larger CFL

time-step than would be permitted by a regular latitude-longitude grid (Li and Saulter, 2014; Saulter et al., 2016). The Arctic
part (cells inside golden circle in Fig. 2) is not used in the operational forecast system at present since for most of the time it
is covered by sea ice.

The GS512L4EUK model is forced by hourly global atmospheric 10m neutral wind files and ice concentration. 10m neutral
winds are provided by a high-resolution atmosphere-ocean coupled global configuration (Williams et al., 2018) of the Unified

Model (UM; e.g., Walters et al., 2019; Brown et al., 2012) and NEMO ocean model each hour. The atmosphere-ocean coupled
model has 0.23° longitude by 0.16° latitude resolution (N1280L70; 2560 latitude x1920 longitude and vertical 70 levels), with
approximately 10km grid length in mid latitudes. Ice concentration is provided by the Operational Sea Surface Temperature
and Sea Ice Analysis (global OSTIA; Good et al., 2020) also produced at the Met Office. Since February 2018 the NEMOVAR
data assimilation analysis scheme is used in OSTIA to combine a background (first guess) ice field with daily satellite ice

information from the Ocean and Sea Ice Satellite Application Facility (product OSI-401-b) (Good et al., 2020). This observed
ice field is gridded using the extended Global ORCA12 grid (ORCA12extL75 tripolar grid at 1/12º resolution with
approximately 9km grid length in mid latitudes and 6km near the UK) and concentrations below 10% are set to zero.
GS512L4EUK uses simple ice blocking (IC0) where grid points covered by ice are treated as land and a cut-off ice
concentration value of 50% at which obstruction begins is used.

GS512L4EUK provides full 2D spectral boundary conditions for the nested operational wave and ocean-wave coupled
configurations. Hence, AS512L4EUK, AMM15SL2 and ocean-wave coupled AMM15 regional models are nested one-way
with lateral boundary conditions supplied from the global wave model (forced by UM neutral global winds at 10km spatial
resolution and 1-hourly) interpolated to the coarsest resolution of the SMC grid (i.e., 25km). Wave boundary data are supplied
as wave spectra onto the outer boundaries.

**2.3.2 AS512L4EUK Atlantic Ensemble Wave Model**

AS512L4EUK is the Atlantic ensemble forecast system for prediction of Atlantic-UK wind waves (Bunney and Saulter, 2015;
Saulter et al., 2016). The grid is based on a cropped version of the GS512L4EUK grid from 25° S to 83° N and provides the
same high resolution around the UK waters, herein, the name convention. Forcing conditions include winds from MOGREPS-
Global atmospheric ensemble and ice concentration from Global OSTIA. MOGREPS-Global is an atmosphere-ocean coupled

model of N640L70 resolution with 1280 latitude x 960 longitude and 70 vertical levels, which is equivalent to approximately
20km grid length in mid latitudes. MOGREPS-Global includes 18 ensemble members with 1 control member and 17 perturbed
members. Post-processing lags the two most recent two cycles to provide probability forecasts from an ensemble of 36
members (34 perturbed + 2 control). Wave boundary conditions at the wet edges where the domain intercepts the Southern
Ocean are provided by the deterministic global wave model GS512L4EUK at 25km resolution. This deterministic boundary

is located sufficiently far south of the north Atlantic storm track that any deterministic swell signal from the south Atlantic is





small compared to the uncertainty in wind-sea/swell systems associated with wave energy generated in the north Atlantic (Bunney and Saulter, 2015). This configuration was first introduced in November 2016 at OS38. Refer to Bunney and Saulter (2015) for a detailed description of the ensemble system.

### 2.3.3 AMM15SL2 UK Waters Wave Model

The UK waters wave model, herein referred to as AMM15SL2, covers the NW shelf-UK area from approximately 45° N 20° W to 63° N 12° E with a resolution of 3-1.5km. The domain extends beyond the shelf break in deep waters in order to avoid boundary issues but focuses on the shelf seas around the UK; i.e., Celtic and Irish Seas, North Sea and English Channel (Fig. 1b). AMM15SL2 configuration name is derived from the AMM15 (1.5km NW Shelf Atlantic Margin Model) ocean model that encompasses the same region and the use of two SMC levels (Saulter et al., 2017; Valiente et al., 2021b). The grid is based
on a two-tier SMC grid refinement (Li, 2011) with variable resolution based on both proximity to coast and water depth. This regional configuration was first introduced in November 2018 at OS40.

The grid resolution is of 3km for water depths larger than 40m and 1.5km for coastal cells with water depths of less than 40m (Fig. 2). The SMC grid is based on a rotated north pole at 177.5° E 37.5° N in order to achieve an evenly spaced mesh around UK. The bathymetry and coastal masking for this configuration is the same as the 1.5km AMM15 NEMO (Nucleus for
European Modeling of the Ocean; Madec, 2008) based ocean configuration (Tonani et al., 2019; Graham et al., 2018). Bathymetry and land-sea mask is based on the European Marine Observation and Data Network (EMODnet portal, September 2015 release) corrected to mean sea level (Tonani et al., 2019). Similar to GS512L4EUK, AMM15SL2 is driven by hourly NWP 10m neutral winds from the global UM-NEMO operational system. AMM15SL2 is also forced by hourly currents from the regional AMM15 Ocean-Wave Coupled Model shelf seas Forecast Ocean Assimilation Model (AMM15-FOAM; Tonani
et al., 2019, see next Sect. for a summary of details).

All forcing conditions in the wave standalone models are interpolated in time and space to the underlying coarsest cell resolution regular grid version of the SMC. This is 25km resolution for winds and ice in GS512L4EUK and AS512L4EUK; and 3km resolution for winds and currents in AMM15SL2.

### 2.3.4 AMM15 Ocean-Wave Coupled Model

The Met Office AMM15 ocean-wave coupled operational forecast system for the NW shelf consists of a WW3 wave model two-way coupled to the ocean NEMO model AMM15-FOAM using the Ocean Atmosphere Sea Ice Soil (OASIS-MCT coupler; Valckle et al., 2015) coupling libraries. The ocean component consists of the NEMO physical ocean model and the Nucleus for European Modelling of the Ocean data assimilation system (NEMOVAR; e.g., King et al., 2018; Waters et al., 2015). NEMOVAR uses a 3D-Var first guess at appropriate time (FGAT) scheme which includes bias corrections scheme for
both sea surface temperature and altimeter data.

Domain, grid and bathymetry of the wave component of this forecast system are the same than for the waves standalone AMM15SL2 UK Waters Wave Model. AMM15 ocean-wave is used in the production of CMEMS Copernicus products for





the NW shelf domain and differs from AMM15SL2 UK waters wave model in the forcing sources. Surface (10m) wind data are provided from the atmospheric high-resolution global configuration of the Integrated Forecast System run at the European

Centre for Medium-Range Weather Forecasts (ECMWF; https://www.ecmwf.int/en/forecasts/documentation-and-support). Temporal resolution is hourly up to T+72 and 3-hourly from T+72 to T+144. The wind field is defined on an approximately 9km resolution. As per the other SMC based configurations, ECMWF wind forcing is pre-processed and interpolated to the level of the coarsest wave model cells, equivalent to approximately 3km. Surface current effects on the waves are included, and surface zonal and meridional current fields from the 1.5km North-West Shelf FOAM-AMM15 model are passed as coupled

fields every hour since OS44. The coupled surface currents are interpolated directly to the SMC grid sea-points, i.e. at the 3km and 1.5km cell resolutions as appropriate (Saulter, 2020b).

### 2.4 Operational production

GS512L4EUK and AMM15SL2 wave models run operationally four cycles per day (00Z, 06Z, 12Z and 18Z; Table 1) to T+66. 00Z and 12Z cycle on each day are extended to a 144hour forecast for the case of GS512L4EUK. Both GS512L4EUK

and AMM15SL2 are initialised using the restart file T+6 from a short 6hour "update cycle" using the most up-to-date NWP winds which include data assimilation.

The operational setup of the wave model does not include data assimilation (Saulter et al., 2020a). However, this is partially accounted for by GS512L4EUK and AMM15SL2 configurations being run in two separate modes: a forecast run and an update run. The forecast suite produces the short-medium range forecasts using NWP forecast winds. This suite also produces the

operational products. The update suite re-runs a short 6hour update cycle using the most up-to-date NWP assimilated winds. This update cycle provides the best possible start conditions for the next forecast cycle. Additionally, ice concentration from global OSTIA for GS512L4EUK and currents from AMM15-FOAM for AMM15SL2 are updated once a day at 00Z. Indeed, AMM15SL2 wave at 00Z runs before the ocean-wave coupled AMM15 00Z in the Met Office production cycle, forcing to use as input file the forecasted currents from the previous day (i.e., currents at T+24 from previous cycle of AMM15 coupled).

AS512L4EUK wave ensemble currently runs as a "lagged" ensemble due to limitations in resource (for both the Atlantic wave ensemble and the driving MOGREPS-Global atmospheric ensemble). At each run cycle, 17 members and the control run to full forecast length with lead times out to 168hour whilst the other 17 members do a short 6hour cycle to maintain continuity. The full-length members alternate at each cycle. Hence, members 0 to 17 run at 00Z whereas members 0 and 18 to 34 run at 06Z and 18Z. A full 36 member lagged ensemble is made up at each cycle from overlapping full length members from the

current and previous cycles.

The AMM15 ocean-wave coupled runs once a day triggering a 144hour forecast. Each model cycle starts with a T-48 hours and does a 2day hindcast prior to each forecast. Initial conditions are taken at T-24. The role of the hindcast is to have a forecast that it is initialised with the best available descriptions for atmosphere and ocean (i.e., with as many observations assimilated as possible). High frequency wind forcing during the hindcast part of each cycle are constructed from the previous ECMWF





analysis cycle. Refer to Saulter (2020a) and Tonani et al. (2021) for more detailed information on the production cycle of this system.

## 3 Model evaluation

Model evaluation is focused on global (GS512L4EUK) and regional (AMM15SL2) UK waters (refer to Fig. 1a,b) configurations. Verification of the systems will be used to document performance characteristics of the baseline configurations,

with a view to further development. For a detailed evaluation of AMM15 Ocean-Wave Coupled Model and AS512L4EUK wave ensemble refer to Saulter (2020b) and Bruciaferri et al. (2021), and Bunney and Saulter (2015), respectively. Performance of the baseline configurations was validated using hindcast analysis experiments (#AN), herein GS512L4EUK-AN and AMM15SL2-AN (Table 2). Trials covered the period from 1st January 2019 to 31st December 2020 and were based on daily cycles of the models forced by NWP 10km resolution operational update cycle winds and the updated fraction of sea ice

(GS512L4EUK-AN) and AMM15 FOAM 1.5km sea (AMM15SL2-AN). The trials were initialised from rest with a 10day spin-up period that was discarded. Each cycle used the restart file at T+24 from the previous cycle. Lateral wave spectral boundary conditions for the AMM15SL2-AN simulation were supplied from the GS512L4EUK-AN simulation. The spectral boundary conditions are provided as external files and interpolated to the 3km resolution outer boundaries.

**Table 2 Experiments specifications for model evaluation.**

| Experiment | Description |
|---|---|
| GS512L4EUK-AN | 2-year (20190101 to 20201231) analysis run global<br>Forcing: Operational archived hourly NWP 10km resolution updated winds and updated fraction of sea ice.<br>Restart at T+24 |
| AMM15SL2-AN | 2-year (20190101 to 20201231) analysis run regional UK waters<br>Forcing: Operational archived hourly NWP 10km updated winds and AMM15 FOAM updated currents.<br>Restart at T+24 |

Wave parameters from the model simulations are evaluated using four different datasets: (i) 6/12-hourly in-situ data from the Joint WMO-IOC Commission for Oceanography and Marine Meteorology's operational Wave Forecast Verification Scheme (Bidlot et al., 2007), herein JCOMM-WFVS; (ii) hourly Daily Ship Synop Observations at fixed platforms across the NW shelf (hereafter SHPSYN); (iii) hourly UK WAVENET and National Network of Regional Coastal Monitoring Programmes (NNRCMP) in-situ observations for coastal waters comprising Waverider buoys (simplified as WAVENET); and (iv) global

satellite merged altimeter data (MA_SUP03) including JASON 2, CryoSat and SARAL AltiKa data. Wind forcing conditions were verified using JCOMM-WFVS, SHPSYN and MA_SUP03 datasets. Wave and wind parameters are assessed for the years 2019 and 2020.

Different wave and wind parameters are evaluated depending on the observation type (i.e., different observation types measure different parameters): significant wave height ($H_s$), mean zero up-crossing period ($T_{02}$) and mean wave direction (Dir) for the





waves and 10m height wind speeds ($U_{10}$) and wind direction ($U_{10}$ dir) for the wind forcing conditions. These parameters are widely used for model evaluation as they give information of the model performance in various aspects. $H_s$ is representative of the bulk energy imparted to ocean surface waves from the atmosphere whereas evaluation of $T_{02}$ helps to assess the ability of the model to represent the wave energy distribution through the frequency domain of the wave spectrum mainly on higher frequencies (i.e., $T_{02}$ is computed using the second spectral moment). Finally, assessment of the wave direction helps to

understand the ability of the model to reproduce the distribution of wave energy in the directional space (Saulter A., 2020b). Basic metrics for model evaluation are described in *Supplement material*. These include *bias*, root mean square deviation (*RMSD*), observations (*SDobs*) and model standard deviation (*SDmodel*), Pearson correlation coefficient (*r Pierson*), standard deviation of the error (*StdE*), covariance (*Cov*) and variance (*Var*). Extreme verification and extra metrics for model evaluation are also provided and include Scatter Index (*SI*) and Symmetric Slope (*SS*) between the model and the observations. *SI* is

calculated dividing the standard deviation of model-observation differences and *SDobs*. The *SS* is described as the ratio of model variance to observations variance.

### 3.1 Wind forcing

Metrics for $U_{10}$ and $U_{10}$ dir against JCOMM-WFVS, SHPSYN and MA_SUP03 observations (Table 3) are computed for the individual domains in order to assess the consistency of these for both configurations. Interestingly, differences in $U_{10}$ metrics

are not significant, indicating that forcing conditions are steady and suggesting that the wind interpolation to the underlaying regular grid with the coarsest SMC resolution (25km for GS512L4EUK and 3km for AMM15SL2) does not degrade the overall wind speed performance. However, $U_{10}$ dir compare closer to observations for the AMM15SL2 domain (*RMSD*=21.49º and 17.00º, *StdE*=21.46º and 16.98º for GS512L4EUK and AMM15SL2 respectively) demonstrating that errors between modelled $U_{10}$ dir and observations are both smaller and more representative of the wind conditions across the NW shelf. This highlights

that wind interpolated to the AMM15SL2 3km retaining the original spatial variability of 10km performs better in terms of $U_{10}$ dir.

Global wind speed ($U_{10}$) verification statistics versus satellite observations (MA_SUP03) are presented in Fig. 3. Underestimation of observed wind speeds by the model occurs in those areas that present either very small or strong mean wind speeds (Fig. 3a,b). Areas which present the lowest mean wind conditions are those across equatorial and close mid

latitude regions during both summer and winter months. These areas present the smallest wind variability (Fig. 3e,f) where overall *RMSD* mean values are of $O.0.6$ ms$^{-1}$ (Fig. 3g,h). Underestimation in these areas seems to be linked to average mean speeds of 5ms$^{-1}$ or lower (i.e., calm wind conditions) possibly also associated to sampling bias from the satellite observations. Very energetic areas such as the Southern part of the Pacific Ocean also present negative bias throughout the year, but these are more exacerbated during JJA months (Fig. 3c; winter in the Southern Hemisphere) during which the largest mean winds

are registered. Equally, negative *bias* is present in the northern part of the Atlantic Ocean also corresponding with the strongest winds (average $U_{10}$>10ms$^{-1}$) during DJF northern hemisphere winter months (Fig. 3d). Positive *bias* during both JJA and DJF always occur in mid latitudes of both hemispheres. This wind overestimation corresponds with modal observed mean wind





speeds (5–10ms$^{-1}$). Largest overestimation rates are observed in those mid latitudes closest to the equatorial strip with negative
*bias*. As expected, model seasonal variability is shown in the verification metrics with largest values of *SDmodel* (>2ms$^{-1}$) and

*RMSD* (1.2ms$^{-1}$) across those areas which present the strongest winds (Fig. 3a,b), southern Pacific Ocean during JJA and
northern Atlantic Ocean during DJF.

**Figure 3.** (a,b) Mean, (c,d) *bias* and (e,f) root mean square deviation (*RMSD*) and (g,h) model standard deviation (*SDmodel*) between
wind (U$_{10}$) forcing conditions and merged altimeter observations (MA_SUP03) across the global domain for GS512L4EUK-AN.
Stats are aggregated every 15-days and averaged for the months June-July-August (JJA; left column) and December-January-
February (DJF; right column).





## 3.2 Wave bulk statistics

The relative importance of wind and current inputs is presented through evaluations comparing GS512L4EUK-AN and AMM15SL2-AN trials against all observations (i.e., WAVENET, JCOMM-WFVS, SHPSYN and MA_SUP03). Summary

statistics for significant wave height ($H_s$) and mean zero up-crossing period ($T_{02}$) and wave direction (Dir) are presented in Table 3. Overall metrics are computed for the individual domains being the entire globe and the NW shelf – UK waters area respectively. Note that evaluation of wave direction (Dir) only corresponds to the coastal waters of the UK.

**Table 3. Summary statistics for wind speed ($U_{10}$), wind direction ($U_{10}$ dir), significant wave height ($H_s$), mean zero up-crossing**
**period ($T_{02}$) and wave direction (Dir): GS512L4EUK-AN and AMM15SL2-AN versus observations of WFVS, SHPSYN, WAVENET and merged altimeter (MA_SUP03) over 20190101 to 20201231.**

| | | GS512L4EUK-AN | | | | | | AMM15SL2-AN | | | | | |
|---|---|---|---|---|---|---|---|---|---|---|---|---|---|
| Variables | Observations | Mean | Bias | RMSD | StdE | SS | r | Mean | Bias | RMSD | StdE | SS | r |
| $U_{10}$ | WFVS | 7.19 | 0.26 | 2.00 | 1.98 | 1.13 | 0.86 | 8.27 | 0.20 | 2.20 | 2.19 | 1.06 | 0.84 |
| | SHPSYN | 8.19 | 0.34 | 1.63 | 1.59 | 0.97 | 0.92 | 8.19 | 0.34 | 1.62 | 1.58 | 0.97 | 0.92 |
| | WAVENET | - | - | - | - | - | - | - | - | - | - | - | - |
| | MA_SUP03 | 7.87 | 0.26 | 1.49 | 1.46 | 1.05 | 0.92 | 8.47 | 0.36 | 1.36 | 1.31 | 1.09 | 0.95 |
| $U_{10}$ dir | WFVS | - | 0.83 | 23.05 | 23.04 | 0.99 | - | - | 0.49 | 14.36 | 14.36 | 1.01 | - |
| | SHPSYN | - | -1.32 | 19.92 | 19.88 | 0.99 | - | - | -1.27 | 19.65 | 19.61 | 0.99 | - |
| | WAVENET | - | - | - | - | - | - | - | - | - | - | - | - |
| | MA_SUP03 | - | - | - | - | - | - | - | - | - | - | - | - |
| $H_s$ | WFVS | 1.88 | 0.05 | 0.29 | 0.29 | 0.95 | 0.97 | 2.08 | 0.09 | 0.28 | 0.26 | 0.95 | 0.98 |
| | SHPSYN | 2.09 | 0.12 | 0.33 | 0.31 | 0.93 | 0.97 | 2.09 | 0.12 | 0.32 | 0.30 | 0.91 | 0.97 |
| | WAVENET | 1.25 | 0.02 | 0.32 | 0.32 | 1.02 | 0.94 | 1.25 | 0.06 | 0.26 | 0.25 | 0.99 | 0.96 |
| | MA_SUP03 | 2.74 | 0.05 | 0.35 | 0.35 | 0.95 | 0.97 | 2.71 | 0.02 | 0.32 | 0.32 | 0.93 | 0.98 |
| $T_{02}$ | WFVS | 6.31 | -0.80 | 1.41 | 1.16 | 0.60 | 0.79 | 6.15 | -0.56 | 0.99 | 0.82 | 0.62 | 0.88 |
| | SHPSYN | 5.98 | -0.58 | 0.99 | 0.80 | 0.71 | 0.86 | 5.98 | -0.56 | 0.98 | 0.80 | 0.69 | 0.87 |
| | WAVENET | 4.52 | -0.24 | 0.78 | 0.74 | 1.19 | 0.86 | 4.52 | -0.12 | 0.67 | 0.66 | 1.15 | 0.89 |
| | MA_SUP03 | - | - | - | - | - | - | - | - | - | - | - | - |
| Dir | WFVS | - | - | - | - | - | - | - | - | - | - | - | - |
| | SHPSYN | - | - | - | - | - | - | - | - | - | - | - | - |
| | WAVENET | - | -0.01 | 32.79 | 32.79 | 0.97 | - | - | -1.55 | 27.58 | 27.54 | 0.97 | - |
| | MA_SUP03 | - | - | - | - | - | - | - | - | - | - | - | - |

Both configurations show good agreement with satellite altimeter observations and in-situ observations of $H_s$. Analysing the observations as a whole (average values), the model and observations present correlation coefficients (Pearson $r$) in the range of 0.94–0.97 and 0.96–0.98 with small *biases* 0.06m and 0.07m for GS512L4EUK-AN and AMM15SL2-AN, respectively. Standard deviations of differences between the models and the observations (*StdE*) are 0.32 and 0.28 which correspond to 13–

25% of the observed mean $H_s$. Mean *SS* are 0.95–0.96 indicating that the *SDobs* is larger than *SDmodel* in both configurations.



Although $T_{02}$ is well reproduced both across the global and the regional domains, mean correlation coefficients are better across the UK waters (i.e., for AMM15SL2-AN): 0.84 for GS512L4EUK-AN versus 0.88 for AMM15SL2-AN. There is a tendency to underestimate observed $T_{02}$ in about -0.54–-0.41s and equally to $r$, mean values of $RMSD$ and $StdE$ are smaller across the UK waters: $RMSD$ = 1.06s and 0.88s and $StdE$ = 0.90 and 0.76 for GS512L4EUK-AN and AMM15SL2-AN,

respectively. Dir metrics show an overall reduction of $RMSD$ and $StdE$ for AMM15SL2-AN (32.8º for GS512L4EUK-AN versus 27.5º for AMM15SL2-AN) suggesting that currents modulation of the wave field and the increased resolution (3km versus 1.5km) help to better capture the wave direction near the coast where bathymetric changes and coastal obstructions are better resolved by the AMM15SL2 configuration. A further contribution to the improved wave direction fields in AMM15SL2 is expected from the wind interpolation as noted in Sect. 3.1.

Model results from GS512L4EUK-AN are compared against satellite observations (MA_SUP03) in Fig. 4. Similar to the forcing conditions, metrics variability match the variability on the wave field. This is: larger values of $bias$, $RMSD$ and $SDmodel$ always correspond with areas with the strongest average wave conditions. Certain $bias$ seasonality is observed with waves underestimated across areas affected by tropical and intra-tropical storms; i.e., tropical, mid and high latitudes in the northern hemisphere during DJF (Fig. 4b,d,f,h) and Indian Ocean during JJA (Fig. 4a,c,e,g). This negative $bias$ during stormy

seasons turns into a positive one of the same order during periods with calmer average conditions (Fig. 4c,d). $H_s$ in other areas, such as the west coast of Australia, appear systematically underestimated ($bias$ = -0.2–-0.4m). The southern part of the South Pacific Ocean shows a large variability in the $bias$ with no clear seasonality, possibly due to cancellation errors. $H_s$ values of $RMSD$ oscillate between 0.1–0.3m in most parts of the globe, with a substantial increase to 0.5–0.6m in those areas with the largest mean wave conditions (i.e., Southern Ocean during JJA and North Atlantic and North Pacific during DJF). This same

pattern is observed in $SDmodel$ (Fig. 4g,h) where larger deviations from the mean values coincide with areas where mean $H_s$ is 4m or larger. Additional large positive biases around island chains and ice edges are also present. It is acknowledged that satellite measurement errors are larger in complex coastlines; however, this overestimation is consistently present throughout the year and is not observed in other coastlines, suggesting that it is more a limitation associated with the model configuration than with observation uncertainties. GS512L4EUK SMC configuration helped reducing biases from previous configurations

(Saulter et al., 2016); however, biases in these areas are still likely due to issues with land/ice masking and the representation of fetch in the model grid.





**Figure 4.** (a,b) Mean, (c,d) *bias* and (e,f) root mean square deviation (*RMSD*), and (g,h) model standard deviation (*SDmodel*) between modelled significant wave height (H_s) and merged altimeter observations (MA_SUP03) across the global domain for GS512L4EUK-AN. Stats are aggregated every 15-days and averaged for the months June-July-August (JJA; left column) and December-January-February (DJF; right column) over 2019-2020.

Model calibration is based on the best overall performance skill. All configurations in the Met Office wave forecast system include the same source term tuning parameters (refer to *Supplement material*) as this has been found to be suitable in previous system versions. Figs. 3 and 4 suggest some imbalance during swell dominated conditions in areas such as the Southern Pacific Ocean and the Indian Ocean where winds over this period appear overpredicted whereas significant wave height is



underestimated (e.g., waters approaching Western Australia; Fig. 4c,d). Something similar, albeit to a lesser extent, occurs in tropical and mid latitudes in the western part of the North Atlantic where, despite a slight overestimation of the forcing conditions, the model shows a negative *bias* with respect to altimeter observations. This imbalance between forcing conditions and model response requires further tuning and/or nesting to improve swell energy propagation from the Southern Ocean for
those specific regions.

Assessment of AMM15SL2-AN modelled $H_s$ and $T_{02}$ against in-situ observations across the UK waters is presented in Fig. 5. Analysing in-situ observations individually allows us to get a more detailed understanding of caveats on the model performance. Hence, following the seasonal pattern observed in the global domain, a weaker performance (i.e., larger values of *bias* and *RMSD*) of the model reproducing $H_s$ is expected when waves are larger (i.e., DJF). Correlation coefficients for $H_s$
are above 0.95 (Fig. 5q,r) and, conversely to the other metrics, $r$ is improved overall during DJF. This improvement in $r$ is even more significant for $T_{02}$ across the North Sea where $r > 0.92$ on average during DJF, versus 0.88 during JJA. Metrics values suggest a good performance of AMM15SL2-AN; however, the model seems to struggle more to replicate the wave energy in the frequency domain during lower energy conditions ($H_s$=1–2m, $T_{02}$=5–6s; Fig. 5a–d). Additionally, $T_{02}$ seems to be consistently underestimated in most locations (*bias*=-0.5s; Fig. 5g,h) whereas $H_s$ is slightly overestimated (*bias*=0.1–0.2m
on average). Although metrics variability between summer and winter months is observed, *bias*, *RMSD* and *StdE* statistics at some specific locations are consistently larger throughout the year (e.g., buoys in the Bristol and English Channels). Whilst the model shows some skill in these regions, the high variability characterised by strong currents due to the tidal range (hypertidal in the case of the Bristol Channel), the fact that those locations are very close to the coast and some local features (e.g., headlands, highly spatially variable bathymetry with features of <3–1.5km spatial scale) are not fully represented by the
model make these regions very dynamic and difficult to resolve more accurately with the current model configuration.

Differences in model skill replicating in-situ observations suggest some platform dependence, especially for the SHPSYN dataset that includes a variety of buoys, lightvessels and fixed platforms. Inspection of the statistics suggests that when using SHPSYN, model versus observations differences increase 10–30% with respect to model comparisons with other observational data and that this is consistent for both model configurations. This suggests that it is not a problem related to the model
performances but with the SHPSYN dataset itself. This issue with the observation quality control procedures has been previously identified in Saulter et al. (2016), where the authors mentioned some metadata inconsistencies such as the sampling time for wave variables and/or type of period data returned by particular platforms.





**Figure 5. (a–d) Mean, (e–h) *bias*, (i–l) root mean square deviation (*RMSD*), (m–p) standard deviation of error (*StdE*) and (q–t) Pearson correlation coefficient (*r*) between modelled significant wave height ($H_s$) and mean zero up-crossing period ($T_{02}$) and in-situ observations across the UK waters domain for AMM15SL2-AN. Stats are computed for the months June-July-August (JJA; left column) and December-January-February (DJF; right column) over 2019–2020. Observations included are JCOMM-WFVS, SHPSYN and WAVENET.**





### 3.3 Comparison of configuration performance

Computation of relative changes in absolute *bias* and *RMSD* as well as differences in model capability to replicate the observations (*Cov, Var* and mean square deviation; *MSE*) is used to assess differences in performance between the different configurations for the NW shelf - UK waters area. Although the different system configurations use the same tuning of the source terms; GS512L4EUK and AMM15SL2 differ in domain resolution (higher resolution for AMM15SL2 configuration) and the inclusion of surface currents as forcing (for AMM15SL2 configuration).

The use of ocean surface currents tends to improve modelled sea states (e.g., Hersbach and Bidlot, 2008; Palmer and Saulter, 2016; Ardhuin et al., 2017; Alday et al., 2021). However, results show that the positive impact of the surface currents and increase in resolution is not always shown in the overall statistics of $H_s$ when comparing the global and the regional configurations. Hence, neutral changes in $H_s$ overall performance exist with 5% increase in *MSE* skill change for coastal areas. Other skill changes in $H_s$ between model baseline configurations are <2% (not shown). A more significant impact is observed

in the mean period with overall improved skill in replicating $T_{02}$ when both currents and a high-resolution configuration are introduced, i.e., AMM15SL2-AN. This is more than 25% and 10% improvement in $T_{02}$ *MSE* and *bias*, respectively, by AMM15SL2-AN with respect to GS512L4EUK-AN (not shown).

The distribution in relative change between AMM15SL2-AN and GS512L4EUK-AN using all in-situ observations in each area of analysis (Fig. 1b) across the UK waters during 2019–2020 is presented in Fig. 6. Following the same pattern as the

overall differences in configuration performance, most of the skill changes in $H_s$ can be considered neutral with some outliers: degradation of $H_s$ by AMM15SL2-AN in *MSE* and *bias* (e.g., around 5% across the Bay of Biscay and 1–2% across the English Channel and the Celtic-Irish Seas). Observed $T_{02}$ is significantly better reproduced by AMM15SL2-AN in all areas of analysis except the North Sea Approaches, where difference in skill change is <-1% (i.e., slight degradation by AMM15SL2-AN). Positive skill changes in $T_{02}$ *bias* and *MSE* for AMM15SL2-AN exceed 25% (improvement) in those areas in the south of the

domain (Bay of Biscay and SW approaches) where there is a combination of off-shelf and on-shelf in-situ observations. It is important to mention that $T_{02}$ observations in these areas are scarcer when comparing with other parts of the domain, which might lead to unrealistic assumptions. Although $T_{02}$ skill change is still positive for AMM15SL2-AN in the other parts of the domain, skill change differences are reduced and oscillate between 2–9%. When analysing relative changes breaking down per location (Fig. 7), those areas where some degradation of AMM15SL2-AN is observed with respect to GS12L4EUK-AN

(Fig. 6) often result from a negative change at a single observations location whilst other locations in the sub-domain see little or no change.



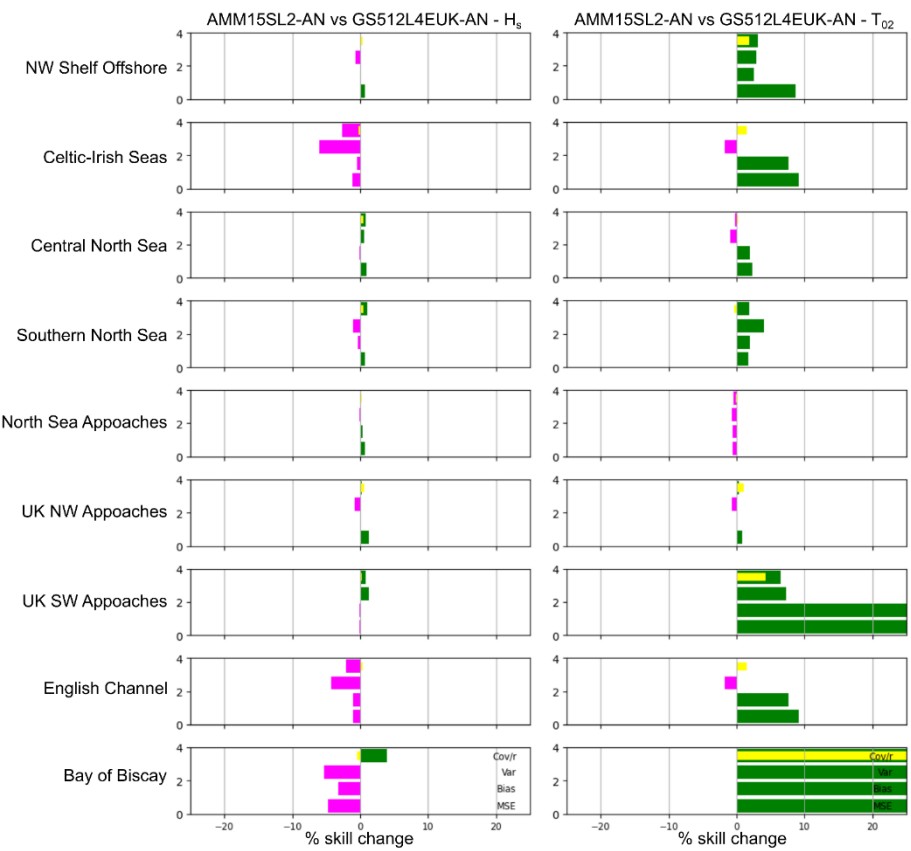

**Figure 6. In-situ observations-model comparison and error changes of significant wave height ($H_s$; left panels) and mean zero up-crossing period ($T_{02}$; right panels) for AMM15SL2-AN versus GS512L4EUK-AN for the different areas of analysis inside the NW shelf – UK waters domain. Magenta (decrease of skill score) and green (increase of skill score) bars represent percent of skill change of AMM15SL2-AN with respect to GS512L4EUK-AN. Yellow bar indicates change in correlation. Refer to Fig. 1 for the extent of the different areas of analysis inside the NW shelf domain.**

Mapping the relative changes between differences in observation-model for AMM15SL2-AN versus GS12L4EUK-AN at each in-situ location (including JCOMM-WFVS, SHPSYN and WAVENET) (Fig. 7) show that main changes in relative absolute *bias* and *RMSD* for both $H_s$ and $T_{02}$ occur at coastal locations. Relative *bias* for $H_s$ varies along the coastal locations indistinctly (increase and decrease), with most of the coastal locations at the SE coast of England showing an increase in the metric for AMM15SL2-AN. Furthermore, $H_s$ *bias* at offshore locations is smaller in most of the locations off-shelf (e.g., SW approaches) with some exceptions (e.g., Celtic-Irish Seas). This improvement in the *bias* for AMM15SL2-AN is also shown in *RMSD* and *StdE* metrics as they tend to be reduced when currents and higher resolution are introduced. Relative changes in $T_{02}$ metrics (improved 9% *MSE*, 3% *bias*, 5% *Var* and 1% *Cov*) show an overall improvement of this bulk diagnostic for AMM15SL2-AN in most locations with some exceptions such as the Scarweather directional wave buoy (Bristol Channel) where, although the tidal modulation of the wave field is only captured by AMM15SL2-AN, it leads to a larger spread on the observation-model differences. This is related to a lag between the model and observations and a potential double penalty effect in the verification



as a result as well as possible cancellation errors in GS12L4EUK-AN. Previous studies demonstrated that metrics based on
direct point matchup between model data and observations might often lead to the double penalty effect (Crocker et al., 2020),
where features are correctly predicted but misplaced with respect to the observations.

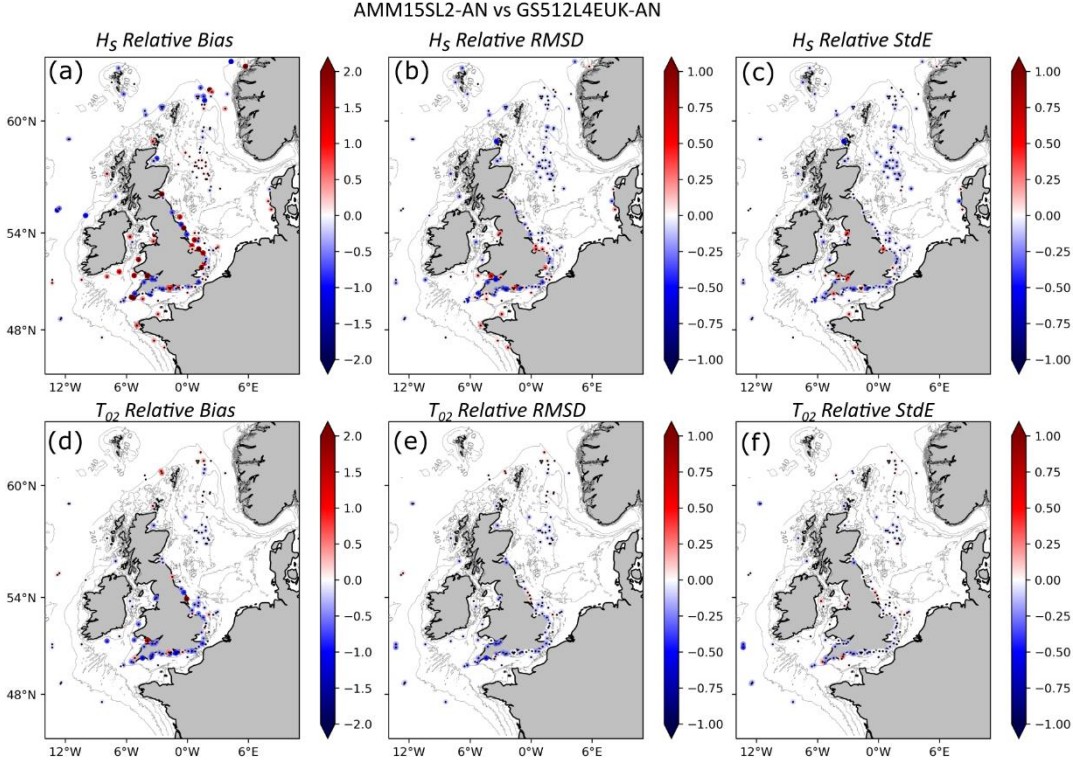

**Figure 7.** Relative change in (a,d) absolute *bias*, (b,e) root mean square error (*RMSD*) and relative standard deviation of error (*StdE*)
between observations-model comparison for AMM15SL2-AN and GS512L4EUK-AN for significant wave height (H$_s$; a-c) and mean
zero up-crossing period (T$_{02}$;d-f) at each in-situ location across the UK waters. Stats are computed over 2019-2020 and observations
included are JCOMM-WFVS, SHPSYN and WAVENET. Negative (positive) values indicate a reduction (increase) of the metric by
AMM15SL2-AN. To facilitate visualisation when no relative change is observed, all in-situ locations are indicated with a black dot.

### 3.4 Waves in coastal waters

Accurate forecast of waves across inshore waters is key in order to reduce damage and disruption by the coast. We emphasise
on the models performance within the coastal areas and evaluate their skill using only WAVENET coastal in-situ directional
wave buoys. It is in the coastal region where noticeable differences between baseline configurations are observed as a result
of increased resolution and tidal modulation only included in AMM15SL2 model. Wave-current interaction in these areas
leads to increased wave height and positive changes in the wave period. Overall, AMM15SL2-AN presents an improvement
in the model *RMSD*, *StdE* and *r* (approximately 5% for H$_s$) respect GS512L4EUK-AN (Table 4) and this improvement is
especially notable for T$_{02}$. In some areas such as Liverpool Bay and Thames Approaches there was no significant change in
the model performance. These correspond to very shallow areas where triad wave interactions become important; however,



this is not included in any of the configurations. GS512L4EUK-AN underestimates $H_s$ in all coastal areas except areas in the E of England (i.e., *bias*=0.11m in SE English Channel and Thames Approaches). This underestimation is not obvious in the overall biases for AMM15SL2-AN and in fact, most of the areas of analysis show slightly overpredicted $H_s$. Conversely, $T_{02}$

is underestimated by both GS512L4EUK-AN and AMM15SL2-AN configurations in all coastal locations but this is significantly reduced for the latter (>15% improvement).

**Table 4. Coastal waters summary statistics for significant wave height ($H_s$), mean zero up-crossing period ($T_{02}$) and wave direction (Dir): GS512L4EUK-AN and AMM15SL2-AN versus observations of WAVENET over 20190101 to 20201231. Refer to Fig. 1b for the areas of analysis. Note that UK Coastal refers to all in-situ coastal buoys.**

| Variable | Area | Mean | GS512L4EUK-AN | | | | | AMM15SL2-AN | | | | |
| | | | Bias | RMSD | StdE | SS | r | Bias | RMSD | StdE | SS | r |
|---|---|---|---|---|---|---|---|---|---|---|---|---|
| $H_s$ | UK Coastal | 1.07 | -0.01 | 0.26 | 0.26 | 0.99 | 0.92 | 0.06 | 0.24 | 0.23 | 1.10 | 0.94 |
| | SW English Channel | 1.11 | -0.06 | 0.25 | 0.25 | 0.99 | 0.94 | 0.05 | 0.25 | 0.25 | 1.11 | 0.94 |
| | SE English Channel | 1.04 | 0.11 | 0.27 | 0.24 | 1.29 | 0.93 | 0.10 | 0.24 | 0.21 | 1.24 | 0.95 |
| | Thames Approaches | 0.89 | 0.11 | 0.25 | 0.23 | 1.49 | 0.90 | 0.13 | 0.26 | 0.23 | 1.45 | 0.90 |
| | Wash Approaches | 1.04 | 0.03 | 0.17 | 0.17 | 0.85 | 0.96 | 0.05 | 0.16 | 0.15 | 0.96 | 0.97 |
| | NE England | 1.17 | -0.04 | 0.20 | 0.19 | 0.80 | 0.96 | 0.00 | 0.18 | 0.18 | 0.81 | 0.97 |
| | Western Isles | 2.88 | -0.03 | 0.40 | 0.40 | 0.83 | 0.98 | -0.03 | 0.37 | 0.37 | 0.83 | 0.98 |
| | Liverpool Bay | 1.12 | -0.03 | 0.17 | 0.16 | 1.01 | 0.97 | -0.02 | 0.18 | 0.18 | 1.11 | 0.97 |
| | Bristol Channel Approaches | 1.50 | -0.15 | 0.34 | 0.30 | 0.81 | 0.94 | 0.02 | 0.26 | 0.26 | 1.04 | 0.96 |
| | Bristol Channel | 0.78 | -0.24 | 0.31 | 0.20 | 0.85 | 0.82 | -0.01 | 0.23 | 0.23 | 1.45 | 0.83 |
| $T_{02}$ | UK Coastal | 4.33 | -0.27 | 0.81 | 0.76 | 1.27 | 0.83 | -0.1 | 0.68 | 0.68 | 1.25 | 0.87 |
| | Southwest Channel | 4.51 | -0.27 | 0.89 | 0.85 | 1.41 | 0.79 | -0.03 | 0.73 | 0.73 | 1.36 | 0.85 |
| | Southeast Channel | 3.99 | -0.16 | 0.68 | 0.66 | 1.41 | 0.78 | -0.06 | 0.60 | 0.59 | 1.34 | 0.82 |
| | Thames Approaches | 3.71 | -0.26 | 0.60 | 0.54 | 1.04 | 0.70 | -0.18 | 0.54 | 0.51 | 0.99 | 0.72 |
| | Wash Approaches | 4.29 | -0.27 | 0.61 | 0.55 | 1.01 | 0.84 | -0.16 | 0.54 | 0.52 | 1.13 | 0.87 |
| | Northeast England | 4.75 | -0.25 | 0.67 | 0.62 | 0.98 | 0.87 | -0.22 | 0.65 | 0.61 | 0.97 | 0.88 |
| | Western Isles | 6.58 | -0.22 | 0.58 | 0.54 | 0.88 | 0.94 | -0.18 | 0.53 | 0.49 | 0.88 | 0.95 |
| | Liverpool Bay | 3.74 | -0.38 | 0.51 | 0.34 | 0.97 | 0.91 | -0.34 | 0.49 | 0.35 | 1.04 | 0.91 |
| | Bristol Channel Approaches | 5.56 | -0.15 | 1.00 | 0.99 | 1.25 | 0.82 | 0.01 | 0.86 | 0.86 | 1.19 | 0.86 |
| | Bristol Channel | 3.86 | -0.81 | 1.23 | 0.93 | 1.08 | 0.41 | -0.34 | 0.87 | 0.80 | 1.25 | 0.60 |
| Dir | UK Coastal | - | 1.13 | 30.46 | 30.43 | 1.024 | - | -0.18 | 23.57 | 23.57 | 1.00 | - |
| | Southwest Channel | - | 3.60 | 26.02 | 25.77 | 1.417 | - | -0.15 | 17.55 | 17.55 | 0.97 | - |
| | Southeast Channel | - | 7.65 | 30.90 | 29.94 | 1.356 | - | 3.46 | 22.47 | 22.20 | 1.07 | - |
| | Thames Approaches | - | -0.55 | 37.74 | 37.73 | 1.487 | - | -2.94 | 32.63 | 32.50 | 1.25 | - |
| | Wash Approaches | - | -0.78 | 37.49 | 37.48 | 1.493 | - | -0.16 | 31.15 | 31.15 | 1.19 | - |
| | Northeast England | - | -2.46 | 33.19 | 33.09 | 1.18 | - | -0.78 | 30.02 | 30.01 | 0.96 | - |
| | Western Isles | - | -11.66 | 28.36 | 25.85 | 1.14 | - | -11.10 | 27.33 | 24.97 | 1.08 | - |
| | Liverpool Bay | - | -7.57 | 24.71 | 23.52 | 1.129 | - | -4.51 | 22.17 | 21.71 | 0.99 | - |
| | Bristol Channel Approaches | - | -2.85 | 21.98 | 21.80 | 1.21 | - | -0.69 | 17.75 | 17.74 | 0.97 | - |
| | Bristol Channel | - | -12.39 | 32.55 | 30.11 | 0.96 | - | -4.30 | 20.08 | 19.61 | 0.80 | - |

The increased resolution in AMM15SL2 (1.5km; Fig. 8) together with the wave-current interactions help to better capture the wave direction near the coast. AMM15SL2-AN shows an acceptable performance in all the coastal areas of analysis. *RMSD* values oscillate from 17º–32º which corresponds to 25% of the observation standard deviation. This percentage in the *RMSD*



increases to 36% for the case of GS512L4EUK-AN. Larger biases for AMM15SL2-AN are found in the Western Isles and are of the order of 11º (Table 4), corresponding to 32% of the *SDobs* (Table 4). Conversely, best metrics are found in the Bristol

Channel Approaches and the SW of the English Channel. These correspond to areas where the tidal range is large but has a maximum of 4–6m and improvements are of the order of 26% respect the other coastal areas.

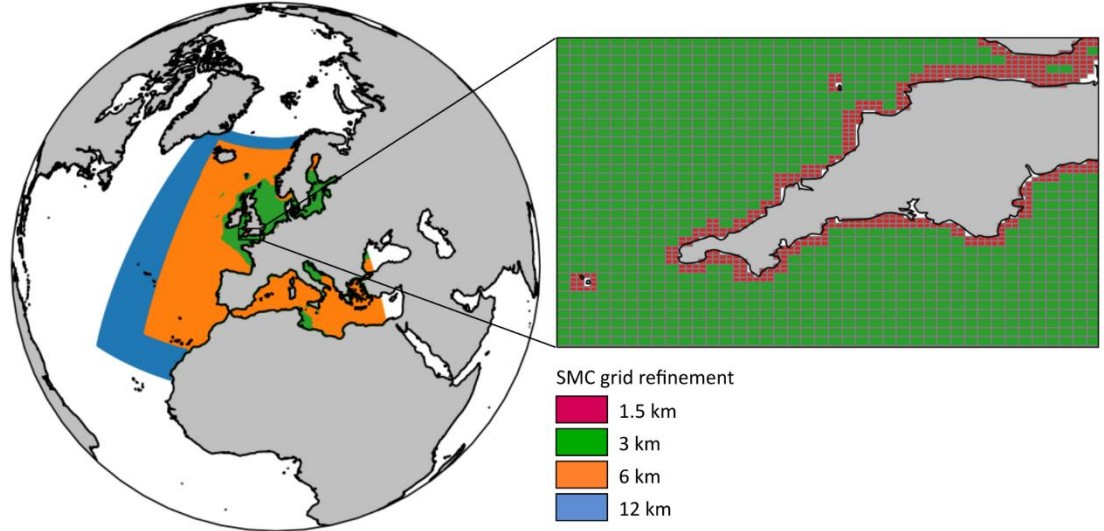

**Figure 8. GS512L4EUK Spherical Multiple-Cell grid refinement (left; highest resolution is 3km) with zoom in the Southwest of England (right; coastal cells resolution is 1.5km) showing coastal waters refinement as resolved in AMM15SL2.**

The improvement of metrics in the coastal areas for AMM15SL2-AN indicates that the model reasonably represents both systematic (long term) and high frequency changes in $H_s$ and $T_{02}$. The addition of wave-current interaction processes in the model results in an incremental improvement in the representation of the wave field. An example of these fluctuations is presented at Start Bay in-situ wave buoy (Fig. 9), a tide modulated coastal in-situ location in the English Channel. Adding the wave-current interactions leads to a reduction of the small negative $H_s$ *bias* at this site from -0.11 to -0.02 m with a reduction

of the *RMSD* from 0.2m to 0.14m. The quantile-quantile relationship for $H_s$ at this location shows that both configurations are in very good agreement with observations ($r$=0.95 and 0.97 for GS512L4EUK-AN and AMM15SL2-AN, respectively) and both tend to underestimate the tail of the distribution; however, this is much closer to observations in AMM15SL2-AN. The $T_{02}$ quantile-quantile relationship shows an overprediction of the lower periods ($T_{02}$=2–6s with *bias*=0.5–1s; Fig. 9c,d) and a significant overestimation of larger periods ($T_{02}$>8s) that is accentuated in the regional configuration. Despite this, $T_{02}$ metrics

are improved in AMM15SL2-AN: *bias* of -0.92 to -0.70, *RMSD* of 1.36 to 1.22s and $r$ of 0.56 to 0.61. In line with the improvement of Dir by AMM15SL2-AN present in most coastal locations, Dir *RMSD* at Start Bay is significantly reduced from 44º to 32.25º and this is reflected in a significant reduction on the model biases. Hence, *StdE* is equivalent to approximately 27% of the *SDobs* at this location.





Figure 9. **(a,b) Significant wave heigh (H$_s$) and (c,d) mean period (T$_{02}$) quantile-quantile relationship and scatter data for GS512L4EUK-AN (left column) and AMM15SL2-AN (right column) at Start Bay in-situ wave buoy. (e,f) Scatter plots for wave direction (Dir) at Start Bay in-situ wave buoy. Inset with wave buoy location is presented in panel (f). Wave bulk stats are included in each individual panel and correspond to the comparison between model and observations over 20190101 to 20201231.**

Tidal modulation of the wave field is observed in several locations. As an example that can be extrapolated to most coastal areas, Fig. 10 shows the timeseries of H$_s$, T$_{02}$ and Dir for both configurations at the Scarweather wave buoy, located in the Bristol Channel, where an increase in the observed T$_{02}$ and H$_s$ can be seen during each tidal cycle. This modulation is only



present in the AMM15SL2 configuration; however, sometimes a lag in the tidal fluctuation (3h for Scarweather; up to 6h in other locations) is present between model and observations that may lead to poorer metrics than when no currents are used (similar to a filtered signal). $H_s$ and $T_{02}$ present an inverse behaviour as the mean period is consistently underpredicted during 510 all stages of the tide but this underprediction is stronger during high tide (Fig. 10b), whereas $H_s$ is overpredicted and this positive *bias* is greater during low tide (Fig 10a). This suggests that other non-linear effects that are important in coastal locations are currently missing in both configurations. Equally, it is also noted the importance of the tidal modulation on the wave direction present in AMM15SL2-AN timeseries captured on the observations within a range of +/-30 degrees (Fig. 10c) in these coastal wave buoys.

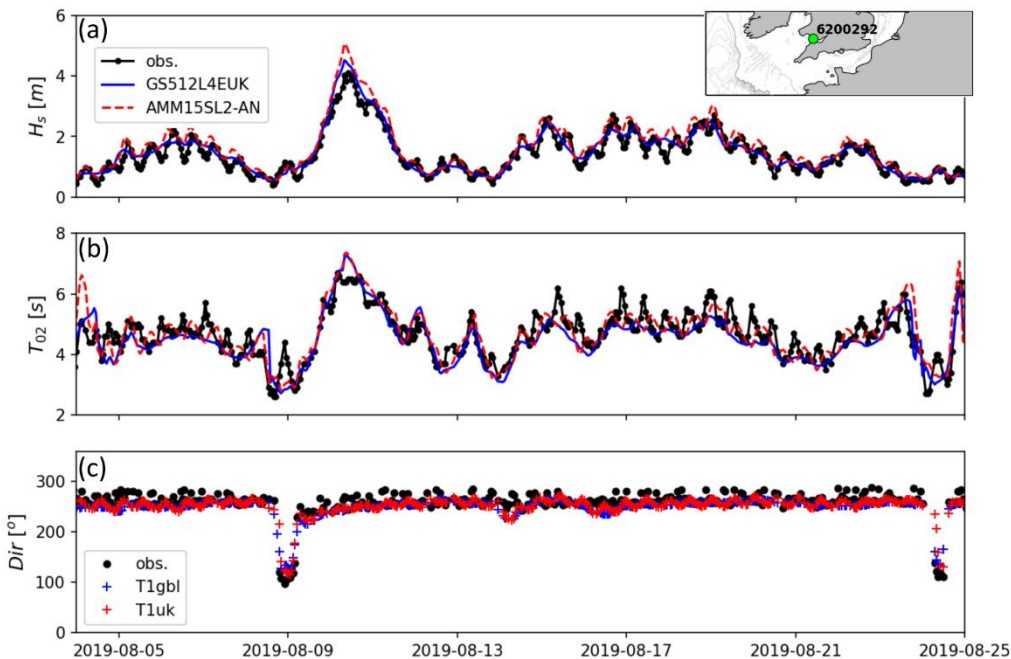


**Figure 10. Timeseries of (a) significant wave heigh ($H_s$), (b) mean period ($T_{02}$) and direction (Dir) for GS512L4EUK-AN and AMM15SL2-AN (right column) at Scarweather in-situ wave buoy. Inset with wave buoy location is presented in panel (a).**

**3.5 Verification of extremes across the NW shelf - UK waters**

Wave models tend to struggle to replicate storm events as uncertainty in both storm evolution in the atmospheric forcing and 520 source term parameterisations is high (Valiente et al., 2021a). Model performance is evaluated for the period of *only storm,* using those observations and model GS512L4EUK-AN and AMM15SL2-AN data exceeding the 90[th] percentile. In this case we define a storm period when $H_s > H_{s,90\%}$ (Q90). Three major features are repeated in the system performance when reproducing the extremes: (i) both configurations tend to overpredict $H_s$ in the sheltered coastal locations and slightly underpredict in other areas of analysis; (ii) $T_{02}$ tends to be underestimated in most of the domain; and (iii) model errors 525 reproducing the wave diagnostics are larger in the SW area of the domain and along very sheltered coastal locations.



There is a tendency to underestimate extremes in most areas of analysis with some exceptions: coastal locations in the E of English Channel and Bay of Biscay. These coincide with locations where $T_{02}$ is also overestimated. Refer to Fig. 10 where the timeseries show how AMM15SL2-AN picks up the peak of the waves and represents the tidal modulation; although sometimes this is translated to larger biases than GS512L4EUK-AN during storms. In other words, the feature of the wave field evolution
may be represented in the model but not always at the exact time in sheltered coastal locations. Equally, during stormy episodes the peak of the wave field is exacerbated with the inclusion of the currents in most coastal locations, sometimes leading to larger overestimation rates than GS512L4EUK configuration; hence, the degradation in *bias* by AMM15SL2 configuration. Conversely, $H_s$ *RMSD* values are reduced in AMM15SL2-AN (i.e., negative relative *RMSD*).

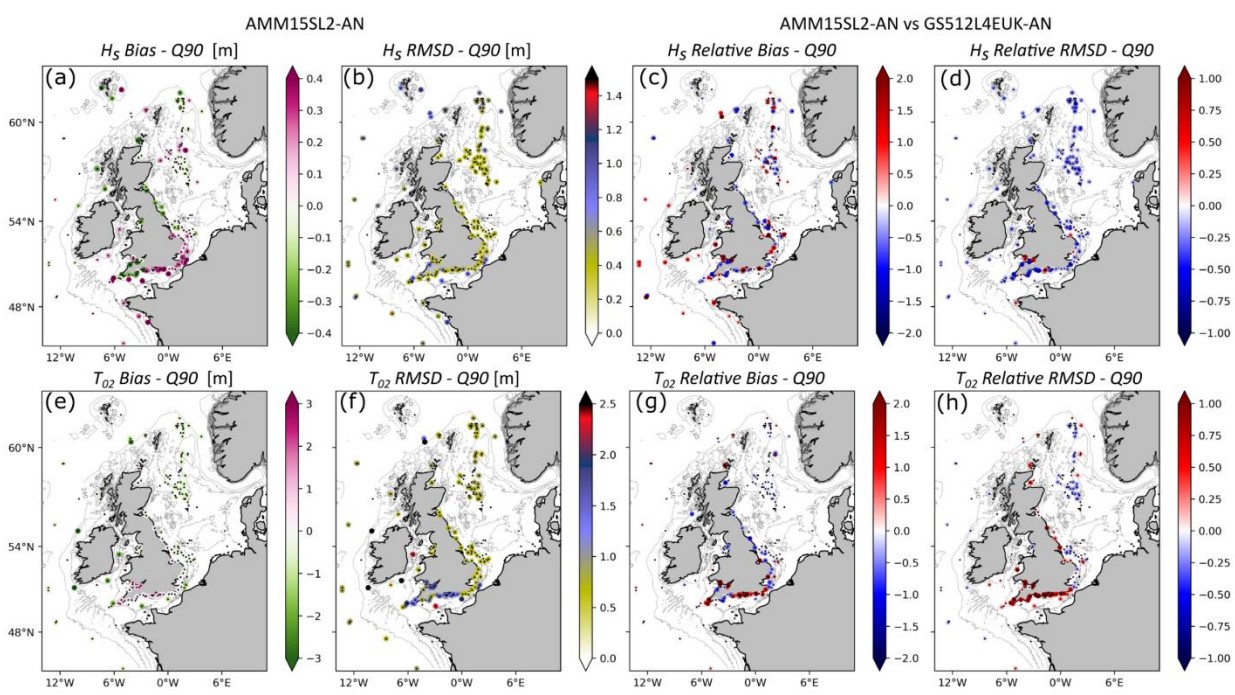

**Figure 11. Observations-model comparison and relative change for the quantile of 90% (Q90) for significant wave height ($H_s$) and mean period ($T_{02}$) in AMM15SL2-AN across UK waters. (a,e) *Bias* and (b,f) root mean square error (*RMSD*) between modelled $H_s$ and $T_{02}$ and in-situ observations; and relative change in (c,g) absolute bias and (d,h) RMSD between observations-model comparison for AMM15SL2-AN and GS512L4EUK-AN. Stats are computed over 2019–2020 and observations included are JCOMM-WFVS, SHPSYN and WAVENET. Negative (positive) values indicate a reduction (increase) of the metric by AMM15SL2-AN. To facilitate**
**visualisation when no relative change is observed, all in-situ locations are indicated with a black dot.**

It should be highlighted that latest model developments improved the system performance during mid to high energy (i.e., extremes) conditions that lead to a significant improvement on the Q90 statistics across the North Sea. Equally, this improvement in representing the extremes was also observed across the coastal locations, mainly for the quantile of the 99% (not shown). Model skill improvement representing the extremes was achieved with the combination of a reduction of the
sheltering for short waves (TAUWSHELTER term) and a bulk adjustment to the wind field through a decrease of the maximum value allowed for wind-wave coupling (BETAMAX term).





### 3.6 Impact of resolution on wave growth

Increased resolution has been demonstrated to play an important role in model skill score. However, advantages on using higher spatial resolution in AMM15SL2 not always show in the overall skills of $H_s$. We know model simulations can show
significant sensitivity to spatial resolution and source term model set-up. In order to test the sensitivity to spatial resolution, we run a number of WW3 idealised numerical experiments with variable resolution. The domain has an extension of 1000km x 500km that is discretised with regular grids of 10km, 5km and 2.5km resolution; experiments 10kmRes, 5kmRes and 2.5kmRes, respectively. All resolutions are then explored for deep water (flat bathymetry of 1000m depth) and shallow water (flat bathymetry of 40m deep) conditions. The model is forced for 48h by a constant wind speed ranging from 10 to 30ms$^{-1}$.
All simulations include the same source term configuration and tuning terms.

Dimensionless fetch limited growth ($gH_s/U_{10}^2$) curves as a function of dimensionless fetch ($gfetch/U_{10}^2$) for the different idealised experiments are presented in Fig. 12. For reference, the theoretical wave growth relationships derived from observations by Young and Verhagen (YV96; Young and Verhagen, 1996) is also included. The difference in wave growth between resolutions is greater for shorter fetches and lower winds. High resolution simulations waves (i.e., 2.5kmRes) generate
higher waves compared to YV96 relationship for both deep (Fig. 12a,c,e) and shallow (Fig. 12b,d,f) water. This behaviour for short fetches is consistent for all wind speeds and differences become smaller for stronger winds. Furthermore, higher resolution presents larger growth rates for shorter fetches and growth differences for the different resolutions decrease with the wind speed. Idealised experiments suggest that the increased resolution in AMM15SL2-AN might lead to faster wave growth and subsequently larger $H_s$ for mid-energy wind conditions in fetch-limited areas. Accordingly, model-observation
results show that for modal conditions, AMM15SL2-AN tends to slightly overestimate $H_s$ and this is better replicated by GS512L4EUK-AN. Conversely, extremes although generally underestimated, tend to be better replicated by AMM15SL2-AN mainly in fetch limited locations. The implication is that in order to obtain a similar behaviour in all model configurations, the next generation of Met Office modelling systems should include a modified parameterization that is domain dependent as the current source term set-up is optimised for GS512L4EUK configuration and modal conditions.

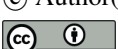

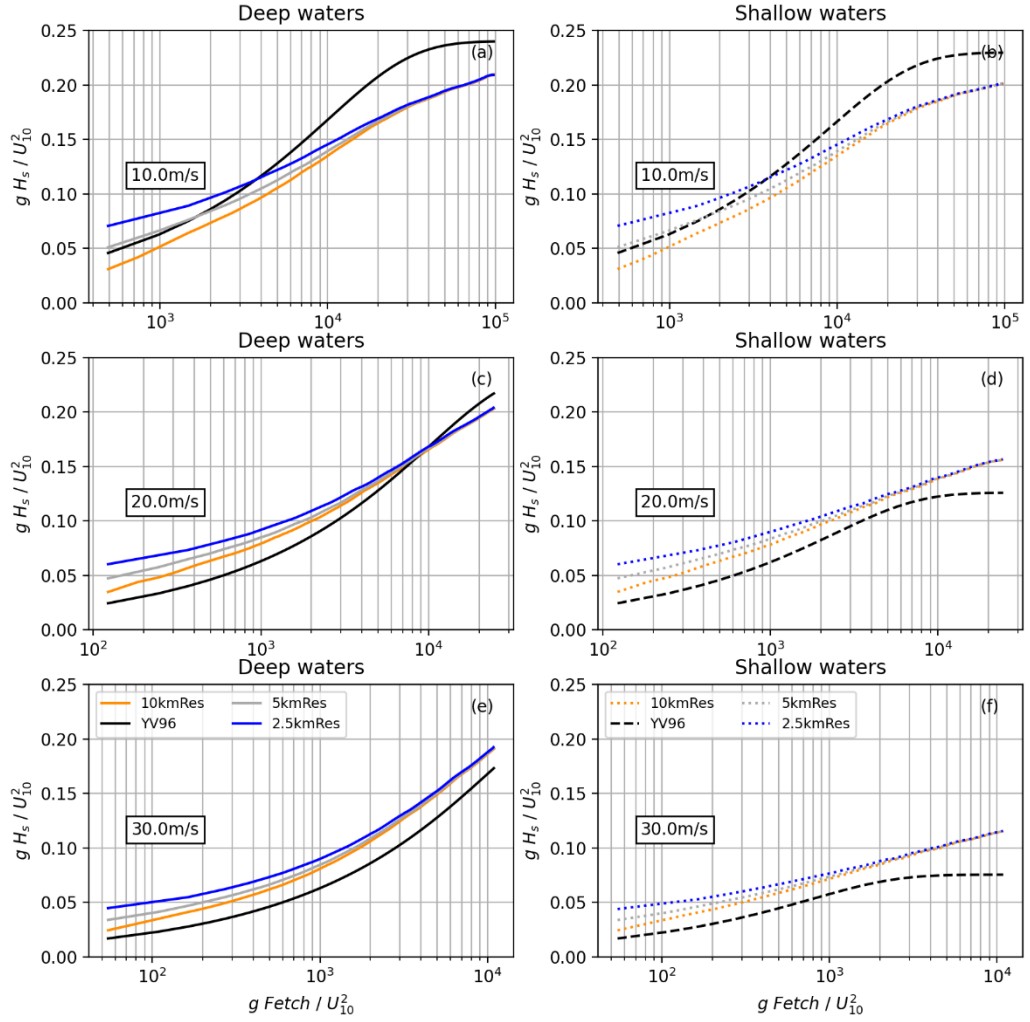


**Figure 12. Dimensionless wave growth curves for different model grid resolutions (10km, 5km and 2.5km) as a function of dimensionless fetch. Fetch limited growth curves are computed for (a,c,e) deep water and (b,d,f) shallow water (40m depth) for constant winds of 10, 20 and 30ms$^{-1}$. The theoretical curve of Young and Verhagen (1996) is presented (YV96). Results for the different configurations correspond to 48h model runs.**

## 4 Forecast performance


Forecast wave system performance for the two baseline configurations was evaluated during 50 days in summer (from 20190619 to 20190814; JJA) and winter (20191204 to 20200124; DJ) across the NW shelf - UK waters. These shorter experiments replicate the latest operational configuration using both the most up to date wave model version (i.e., WAVEWATCH III® at version 7.12) and the most up to date NWP winds (as described in *Sect. 2.4*). FCST experiments used


the corresponding T+6 restart output file generated in the analysis runs for initialisation of each trial. All FCST runs cycled every 6-hours with the 00Z cycle on each day triggering a 144-h forecast. Hence, updated winds were used for the first six





hours of each cycle adding the forecast winds after the first six hours of 00Z cycle. Refer to *Sect. 2.4* and Table 1 for more information on the temporal and spatial resolution of the forcing conditions. For comparison purposes, AMM15SL2-FCST runs was up to T+144 as per GS512L4EUK-FCST, as opposed to T+66 used in operations. It is noted that currents as forcing
were lacking in the last 78 hours of the AMM15SL2-FCST runs.

**Table 5 Experiments specifications for forecast capability.**

| Experiment | Description |
|---|---|
| GS512L4EUK-FCST | JJA (20190619 to 20190814) and DJ (20191204 to 20200124) forecast run global. Forcing: forecast 10km NWP winds and updated fraction of sea ice.<br>Restart at T+6<br>T+144 forecast at 00Z cycle |
| AMM15SL2-FCST | JJA (20190619 to 20190814) and DJ (20191204 to 20200124) forecast.<br>Forcing: forecast 10km NWP winds and AMM15 FOAM analysis and forecast currents.<br>Restart at T+6<br>T+144 forecast at 00Z cycle |

Fig. 13 shows bias and RMSD for wind forcing conditions over the summer months of JJA and the winter months of DJ. Forecast evaluation encompasses from T+24 hours to T+144 hours. Winds tend to be overestimated in both configurations during most of the forecasting period up to T+96. Further inside the forecast lead time, winds appear to be slower versus the
first forecast days, and the tendency is to show a reduced *bias* that might be also associated with cancellation errors (Fig. 13a,b). This overall wind speed decrease is more accentuated over the winter months (DJ) in both systems. GS512SL4EUK and AMM15SL2 *biases* are almost constant around 0.4–0.6 and 0.1–0.3 ms$^{-1}$, respectively up to T+96, decreasing to 0.1 and - 0.2 respectively at the end of the forecast. $U_{10}$ *RMSD* oscillates 1.5–2.5ms$^{-1}$ up to T+96 and increases to 3.5ms$^{-1}$ in the last two days of forecast. *RMSD* for $U_{10}$ dir is almost equal for the two systems during both winter and summer months with values
oscillating from 20° at T+24 up to 60° at T+144 (Fig. 13g,h). This indicates that the errors between model and observations are steady in space and increase in time.



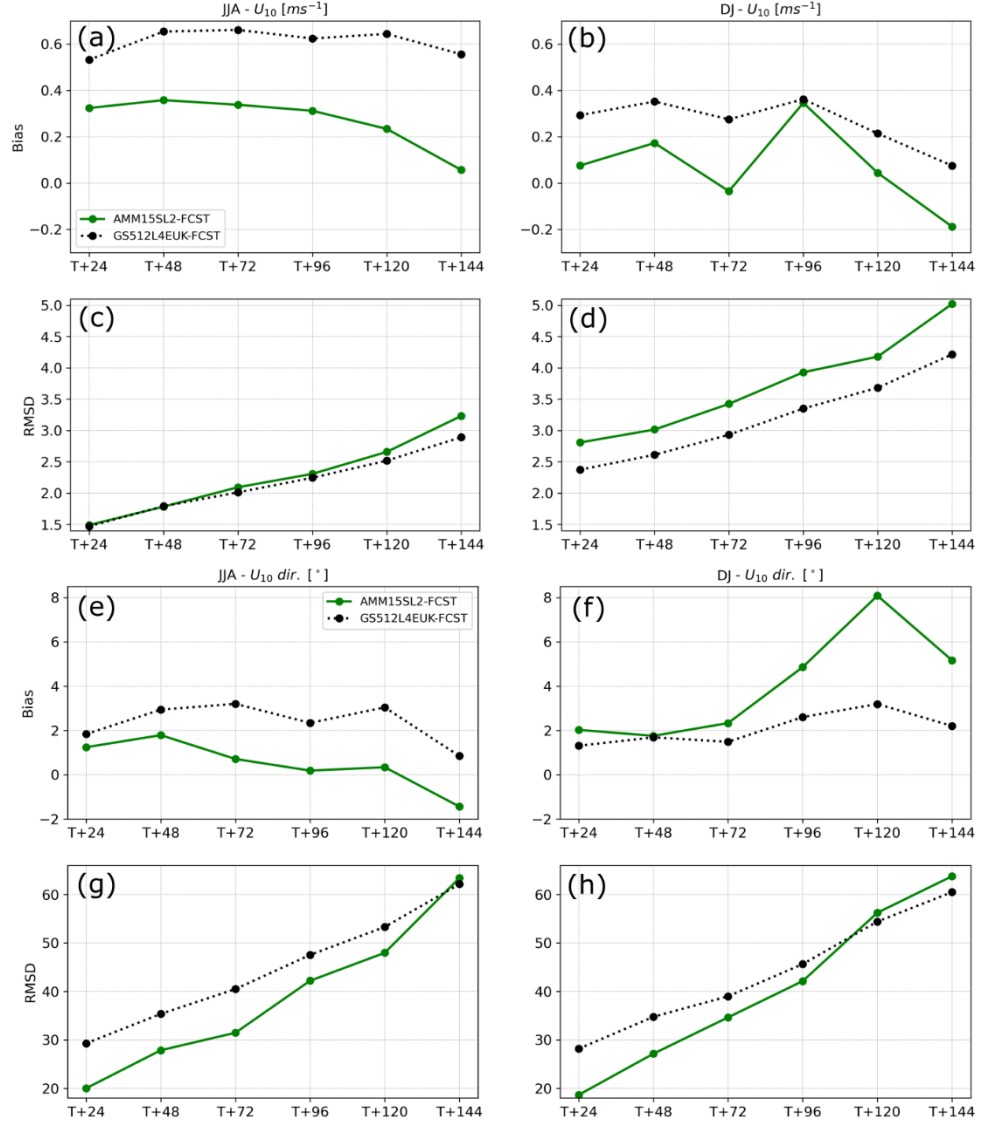

**Figure 13. Forecast (a,b,e,f) *bias* and (c,d,g,h) root mean square deviation (*RMSD*) for wind speed (U₁₀; a-d) and wind direction (U₁₀ dir; e-h) forcing every 24 hours over a forecast period of 6 days (T+144). Values are averaged over the months June-July-August (JJA; left panels) and December-January (DJ; right panels) and correspond to the NW shelf – UK waters model (AMM15SL2-FCST; solid line) and the global model (GS512L4EUK-FCST; dashed line).**

Forecast skill of wave parameters over the summer months of JJA and the winter months of DJ is presented in Fig. 14. As demonstrated in the hindcast runs, the impact of the surface currents is not always shown in the overall statistics of H$_s$. Indeed, the larger spread on the observation-model differences in AMM15SL2-FCST due to the tidal modulation leads to a small degradation in the model *biases* for the FCST run respect the global configuration. H$_s$ *bias* for AMM15SL2-FCST during the forecast period is greater than for GS512L4EUK-FCST (Fig. 14a,b) whereas differences in *RMSD*, although still larger for AMM15SL2 overall, are smaller (Fig. 14c,d). Consistently with the #AN runs, AMM15SL2-FCST shows a better performance





with >20% reduction in *RMSD* compared to the global configuration for $T_{02}$, due to better representation of bathymetric features, depth related processes and wave-current interaction (Fig. 14e-h). Metrics seasonality for both $H_s$ and $T_{02}$ bulk
parameters is also observed in all the leading times with larger values of *RMSD* during DJ.

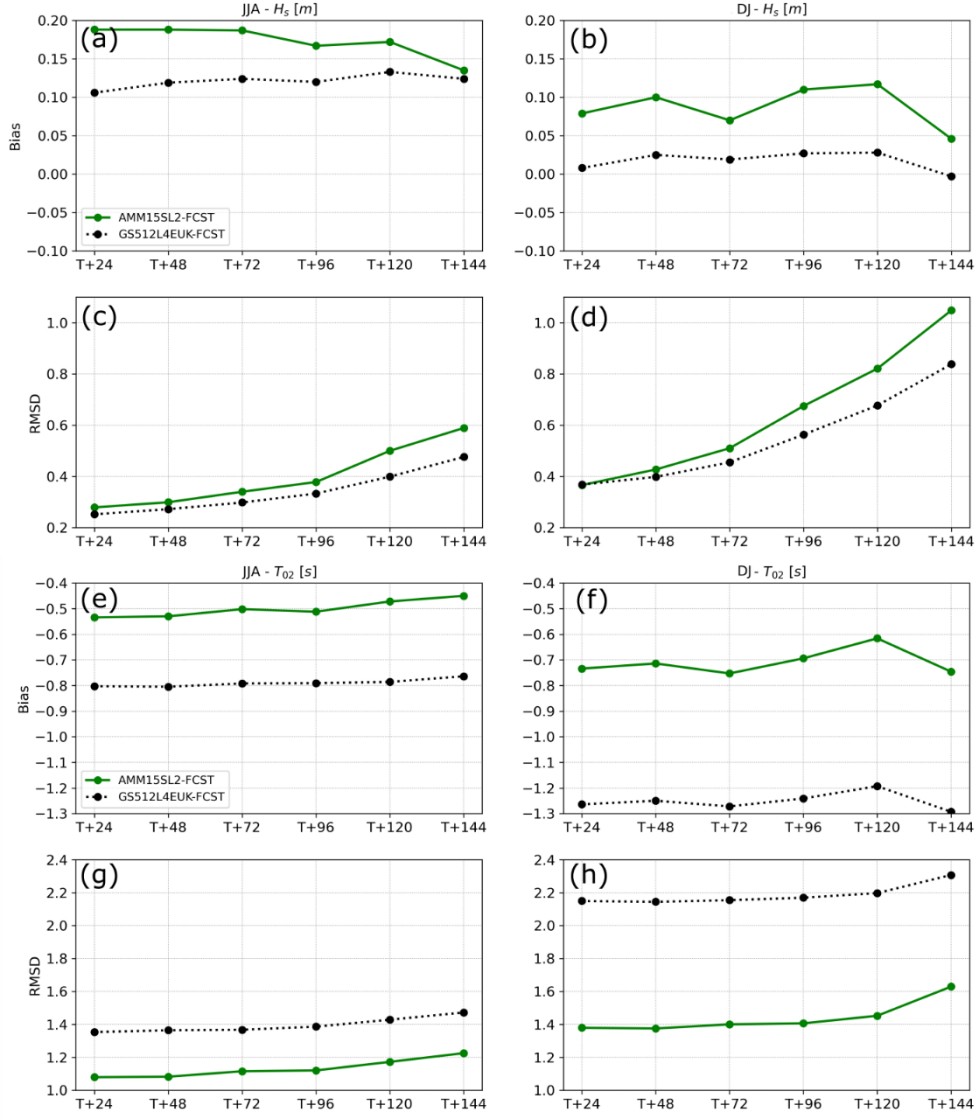

**Figure 14. Forecast (a,b,e,f)** *bias* **and (c,d,g,h) root mean square error (*RMSD*) for significant wave height ($H_s$) and mean period ($T_{02}$) every 24 hours over a forecast period of 6 days (T+144). Values are averaged over the months June-July-August (JJA; left panels) and December-January (DJ; right panels) and correspond to the UK waters model (AMM15SL2-FCST; solid line) and the global**
**model (GS512L4EUK-FCST; dashed line).**

As expected, the forecast skill of both configurations decreases steadily with lead time (i.e., positive trend) for both forcing and wave bulk parameters (i.e., $U_{10}$, $H_s$ and $T_{02}$). This decrease in the forecast skill appears to be relatively steady for the first four days of forecast (up to T+96); however, *RMSD* trend indicates a more rapid decrease in the forecast skill after these. It is


noted that the degree of decrease in the forecast skill for the case of $T_{02}$ is smaller comparing with $H_s$ and in fact, values of
*bias* (-0.8s and -0.5s for GS512SL4EUK-FCST and AMM15SL2-FCST during JJA, respectively) and *RMSD* (1.4s and 1.1s
for GS512SL4EUK-FCST and AMM15SL2-FCST during JJA, respectively) are almost constant for the first four days in both
JJA (Fig. 14e,g) and DJ (Fig. 14f,h) periods.

## 5 Summary and future developments

The latest developments and performance of the current Met Office operational wave system have been presented. Performance
of the system was focused on the global (GS512L4EUK) and the NW shelf - UK waters (AMM15SL2) baseline configurations.
The evaluation of system performance and forecast capability shows a good agreement with both satellite and in-situ
observations and demonstrates the quality and accuracy of the Met Office wave forecast capability. As expected, forecast skill
decreases steadily with lead time for both forcing and wave parameters; however, this decrease is slower up to T+96. Model-
observations correlation is beyond 0.94–0.96 in all areas of analysis with standard deviations of differences that correspond to
maximum 13–25% of the observed mean bulk wave diagnostics. The inclusion of wave-current interaction and the higher
resolution for depths <40m in AMM15SL2 together with a better representation of the local features (e.g., headlands, highly
spatially variable bathymetry) help to significantly improve the prediction of the wave direction near the coast within a range
of +/-30 degrees and the mean period with >20% reduction in the *RMSD*. This is also a consequence of the increase in wind
forcing resolution (10km). Winds in AMM15SL2 are not up-scaled in the pre-processing routine as it is performed in
GS512L4EUK (i.e., 10km resolution winds interpolated to a 25km regular grid). $H_s$ is better forecasted overall on the
GS512L4EUK, which we attribute to an energetic tunning of the source term yielding in an overestimation on AMM15SL2;
consequently, extremes are better forecasted on the AMM15SL2. This improved skill, together with a better prediction of
mean upcrossing period and wave direction, has large implications for the prediction of waves approaching coastal locations
and subsequently in beach safety, risk to flooding and overtopping and shoreline evolution in general. It is also recognised that
despite a good skill of AMM15SL2 replicating inshore waves, the model utility in coastal zones largely sheltered and/or with
strong shallower water bathymetric variability should be treated with caution as there are important non-linear effects that are
not included in any of the baseline configurations.

Recent studies have demonstrated positive impacts on significant wave height prediction when surface ocean currents are
accounted for (e.g., Hersbach and Bidlot, 2008; Palmer and Saulter, 2016; Ardhuin et al., 2017; Echevarria et al., 2021;
Valiente et al., 2021b). AMM15SL2 based configuration includes wind and sea surface currents as forcing conditions. Accurate
representation of the wave-current interactions across the NW shelf - UK waters domain is essential as ocean-wave coupling
improves accuracy of the ocean surface dynamics by 4% (Bruciaferri et al., 2021). Additionally, it is clear that the presence of
currents can modify the distribution of the wind waves on the shelf with >15% impact during modal conditions (e.g., Ardhuin
et al., 2012; Valiente et al. , 2021b; Alday et al., 2022). The quantitative assessment to demonstrate the improvement of the
forecast skills in the significant wave height diagnostic by AMM15SL2 respect GS512L4EUK has been proved difficult. A





lag between model and observations is present in some in-situ locations that leads sometimes to poorer metrics than when no currents are used (i.e., GS512L4EUK). Although relative changes in $T_{02}$ metrics and wave Dir show an overall significant improvement (>25% in *RMSD* and 10% in *bias*) of these diagnostics when currents and higher resolution are introduced, a larger spread on the observation-model differences for $H_s$ is also observed. This is considered to be linked to the double penalty

effect (Crocker et al., 2020), where features are well predicted but misplaced with respect to the observations.

Imminent system developments include: (i) the use of sea point wind forcing in the SMC grid, improving the wind transfer between atmosphere and ocean; and (ii) the optimisation of the models in line with model resolution. The change in the pre-processing of the wind forcing conditions task will include sea point winds (i.e., SMC grids cells) instead of the current pre-processing step where winds are interpolated to the underlaying grid resolution (25km for GS512L4EUK and 3km for

AMM15SL2) in which 10km NWP Met Office wind resolution for the global domain is up-scaled. This development will help correcting some of the wave model behaviour in certain areas of the globe where an improvement in wind speed and direction due to the higher resolution interpolation is likely to be an important factor. Additionally, idealised scenarios showed resolution dependent wave growth indicating that it is important to optimise the source term parameterisation for the different spatial resolutions. Model-observation errors observed in AMM15SL2 for modal conditions are expected to be reduced after the

retuning of the regional model to better match observations across the coastal UK waters as currently this is more optimised to better capture extremes and for the global model.

Long term improvements in the Met Office operational wave forecasting system will focus on various areas which include improving computational efficiency with the use of the SMC multigrid and exploring data assimilation. The Met Office has recently demonstrated the benefits of running SMC in multigrid mode (Li, 2022) and the next development steps will be to

work towards implementing a multigrid approach for the global domain that will allow for a hybrid parallelisation (component and domain decomposition) in hybrid MPI-OpenMP mode. This improvement was tested and results show large reductions in computational time and memory demand, permitting future model updates with increasing resolution (Li, 2022) and a more efficient use of high-performance computers based on GPUs. Additionally, various studies have shown benefits from using data assimilation to improve the wave initialisation (Saulter et al., 2020a, Aouf et al., 2021). Previous attempts using

AMM15SL2 configuration demonstrated a small positive impact of the assimilation scheme on $H_s$ forecast skill over lead times of up to 12h. In future years we will explore the benefits of using data assimilation in our operational systems.

Future systems will include the waves as a system component of a more comprehensive atmosphere-wave-ocean-land-ice system. Met Office research is also focused on delivering more accurate and comprehensive forecasts of the wider earth system (Graham et al., 2018; Tonani et al., 2019). This implies, in most cases, a need to develop more integrated systems where the

different physical components (i.e., atmosphere, ocean, ice and waves) are coupled (e.g., Lewis et al., 2019; Bruciaferri et al., 2021; Valiente et al., 2021a; Castillo et al., 2022). In recent years, the Met Office has put significant effort into the development of fully coupled (atmosphere, wave, ocean and sea-ice) global and regional models and although AMM15 ocean-wave coupled system has been released, other coupled systems are still far from becoming operational. The GS512L4EUK wave model described in this paper is in the process of being implemented in a global research atmosphere-ocean-ice-wave coupled





configuration; however, it will need time before it becomes operational. Additionally, these coupled systems are often more complex and computationally expensive with increased resource demands over a traditional standalone model. Met Office internal testing demonstrates that a coupled simulation increases 10% the running time per model respect their standalone version; i.e., if an ocean model needs $n$ nodes to run and a wave model needs $m$ nodes, the ocean-wave coupled simulation of the two will need $n+m$ nodes with an increase of 20% in the running time. While studies continue toward a fully coupled

prediction system with atmosphere, ocean, land, ice and wave components, the maintenance and development of each of the model components is crucial in NWP.

*Data availability.* The length, resolution and spatial coverage of the data generated in running the trials described in this paper requires a large storage facility. The complete or partial data will be available via request to the corresponding author. Data

used for the model evaluation and analysis in this paper in the form of model-observations match-up netCDF files are available via https://doi.org/10.5281/zenodo.7019826.

Datasets for model evaluation include different sources. SHPSYN in-situ observations is accessed via the Global Telecommunication System but it is also publicly available via http://www.marineinsitu.eu/dashboard/. WAVENET in-situ data is obtained from the National Network of Regional Coastal Monitoring Programmes and CEFAS Wavenet, and should

be available prior registration at http://www.channelcoast.org/ and https://www.cefas.co.uk/cefas-data-hub/wavenet/. JCOMM-WFVS observations are obtained as Met Office is part of the World Meteorological Organisation - International Oceanographic Commission (WMO-IOC) Joint Commission On Marine Meteorology's operational Wave Forecast Verification Scheme. MA_SUP03 satellite altimeter data is available for public download and can be obtained prior registration via FTP in ftp://ftp.ifremer.fr/ifremer/cersat/products/swath/altimeters/waves/data .

Additional information on the data acquisition of the different observational datasets used in this paper is included in the *Supplement material*.

*Code availability - Obtaining WAVEWATCH III[®].* The version of WAVEWATCH III used operationally at the Met Office is publicly available via the Met Office's "Trusted Institutional Fork" of the NOAA WW3 GitHub repository:

https://github.com/ukmo-waves/WW3/tree/ukmo_ps45-1.hotfixes. The WAVEWATCH III code base is distributed by NOAA under an open-source style licence via http://polar.ncep.noaa.gov/waves/wavewatch/wavewatch.shtml (NOAA, 2021a). Interested readers wishing to access the code are requested to register to obtain a license via http://polar.ncep.noaa.gov/waves/wavewatch/license.shtml (NOAA, 2021b). Refer to Supplement material for more details.

*Code availability - Obtaining configuration files.* Basics of the system configuration including grid, modules and tuning parameters files are publicly available via https://doi.org/10.5281/zenodo.7148687.





*Author contributions.* N. G. V. conceptualised the experiments, set-up and run baseline trials, conducted the formal analysis and wind and wave verification as well as data curation, wrote the original draft and prepared figures/visualization; A. S. and B. G. helped with conceptualisation, validation resources and review and editing of draft; C. B. and J. L. contributed to model description and provided a figure for SMC visualisation; C. P. performed AMM15 currents validation; C. B., A. S., J. L., N. G. V. and T. P. contributed to the development of the wave operational system. All the co-authors contributed to the edition of the final version of the manuscript.

*Competing interest.* The authors declare that they have no conflict of interest.

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
