# Peer review of "The Met Office operational wave forecasting system: the evolution of the Regional and Global models"

_Geoscientific Model Development, 2022_

## Author Comment (AC1)

**Response to Referee #1 Comments**

Anonymous Referee #1
Referee comment on "The Met Office operational wave forecasting system: the evolution of the Regional and Global models" by Nieves G. Valiente et al., Geosci. Model Dev. Discuss., https://doi.org/10.5194/gmd-2022-261-RC1, 2022

Summary
The paper describes and evaluates four wave models run by the UK Met Office.

Our thanks to the reviewer for some constructive critiques. Comments have been addressed and we feel these have significantly improved the manuscript.

The manuscript has been modified substantially and special emphasis has been put in: (i) reduction of content, (ii) restructuring of manuscript, and (ii) highlight of main contributions that will lead to further numerical model development. The latter has paid particular attention to the influence of spatial resolution mainly in coastal settings and effects of the tidal currents on the wave field. We believe these outcomes will contribute to further develop not only the Met Office system but will potentially help with the optimisation of other wave systems.

Note that line number corresponds to the revised manuscript with "All Markup".

General comments
The manuscript would be excellent as a technical report, but has been submitted as a journal article, so I review it in that context. It is not yet suitable as a journal article. It is too long, and too much of the content will not be of much interest to readers beyond the author list.
I also hope that the authors recognize that many of us (S&E professionals) spend a lot of time writing, but not everything we write should necessarily be submitted for peer review. Some can be unpublished. Some can be published in technical reports. Some can be submitted for peer-review. As authors, we need to down-select and curate, and we should be careful not to impose unnecessary burden on the volunteer review system.

As both authors and reviewers we appreciate the time that has been taken by all the reviewers of this manuscript to give it a fair reading. Our intent in the original manuscripts was to follow GMD's guidance that a model description paper should include a comprehensive description of the numerical model if this falls within the scope of the journal. It must describe both the underlying scientific basis, purpose of the model and give an overview of the numerical solutions employed in order to be easily reproduced. Additionally, "all technical details which could substantially affect the numerical output should be described". However, the multiple component nature of the Met Office global-regional and operational numerical wave modelling made for a long manuscript in order not to compromise reproducibility aspects.

The manuscript describes the current Met Office WW3 based wave operational forecasting system with unstructured multi-resolution. The data produced by these models support national and international business, as well as governmental and research activities that require good quality forecast to complete their functions. In particular, the models described in this paper are currently being used in several international projects for climate projections (e.g., used to generate boundaries for CHAMFER, see https://noc.ac.uk/projects/chamfer; global and regional configurations are run as part of the Met Office-NOC effort for wave climate projections using UKCP18 winds). In this regard, we felt it was essential to document the system and those new developments that have not been addressed in a peer review publication to date and could potentially reach a wider audience in the wave modelling community. The significant number of views and downloads of the preprint also supports this idea and suggests that there is some wider interest in a description paper documenting the Met Office wave operational system.

Following the reviewer comments, we have focused the restructured manuscript on the key elements that make this set of configurations unique (e.g., SMC multigrid resolution) while highlighting those outcomes that although particular to the Met Office configurations, offer some insights for the optimisation of other wave systems. We hope these changes address the reviewers, and we believe that the amendments we have made to the manuscript have helped to highlight the main contributions that were possibly hidden in too much detail in the previous manuscript version.

The paper describes and evaluates four different models. This is too ambitious for a paper, unless it is a glossy "overview" type of paper.

We agree. Although some references to all the models that are part of the operational system have been left, these have been kept to the minimum and the manuscript now focuses on the two baseline configurations: global GS512L4EUK and regional AMM15SL2. Most results regarding influence of resolution and wave-current interactions in model performance can be extrapolated to the other two configurations that have been left out.

The abstract is too long.

We agree. The abstract has been reduced by a 15%. Please refer to L7-21.

The last section is too long. The reader should be able to find quickly what has been accomplished in the paper, without having to sort through a lot of discussion. Discussion and conclusions should be separated, and the conclusions should be a bald summary of what was found/accomplished.

Discussion was trimmed and is now "Discussion and ongoing development" (L474-531). This section is now similar to other examples in GMD (e.g., Lewis et al., 2019; Castillo et al., 2022). As suggested by the reviewer, "Conclusions" (L532-561) were added as an individual section where the main outcomes were lined up.

The English is excellent grammatically, and the paper is well-proofed (I found only one typo) but the writing appears rushed and haphazard in places. The sentence connectors do not always work (e.g. see how the word "hence" is used). There are references to things that are not explained (and should be explained or not mentioned). And, paradoxically, too much detail in many places, with much of the text unbearably laden. Lines 245-250 are a good example.

We hope that both the new manuscript structure and the focus on the wave-current interaction and resolution in the accuracy of the models helps build a story that is now not rushed and more clearly explained. The main authors went through the text and have worked to present this more fluently.
L245-250 text referred to the ensemble configuration. Those lines were removed from the manuscript as now it focuses on the baseline configurations GS512L4EUK and AMM15SL2.

I recommend that the authors start with a list of interesting and novel findings from their work and use that as a basis for the paper. It may help to think about how much time an average reader will devote to reading the paper. One hour? Two? Start there, and then consider how to best present the most useful information. All six co-authors should read the manuscript carefully and think about how the text can be more considerate of the readers' situation. Find ways to draw attention to the most useful parts, so that they are not missed by a reader who only wants to spend a very short time with the paper (e.g. 10 minutes).

Thanks to the reviewer for these recommendations. Following this approach, we have completely restructured the manuscript. This also led to a significant reduction of the content. We have now put special focus on the impact of resolution and the effect of wave-current interactions in the model accuracy. We believe that the addition of the conclusions where the main outcomes are highlighted will contribute to further develop and optimise not only the Met Office forecasting system but other wave numerical models.

I estimate that the paper is between 13000 and 13500 words from abstract through conclusions. I suggest that 8500 words be used as a upper limit, though of course, a good paper could be shorter or longer based on the quantity of interesting findings and useful information.

We do appreciate the extension of the manuscript, and there has been an 20% reduction in the number of words without, we believe, compromising the description of the models and level of information needed for reproducibility purposes. Operational models are subject to rigorous evaluation that include spatial verification over long periods of time against various observation datasets, focus on locations for certain situations and evaluation of general model properties. All these aspects are covered in sections 4 and 5 and we believe those are necessary to provide a comprehensive picture of the model behaviour. We do appreciate these have an impact into the extension of the manuscript but hope all the discussions included are meaningful to describe different aspects of the model performance.

The model description (section 2.2) needs to be checked by someone that is deeply familiar with the model features being described. I got the impression that the writer of this section is very familiar with the model's technical implementation and operation but less familiar with at least some of these features.

Thanks for the comment. This section has now been checked and modified accordingly by two of the co-authors who are very familiar with WW3 model source terms and parameterisations. Refer to L71-126.

---

## Author Comment (AC2)

**Response to Referee #2 Comments**

Anonymous Referee #2
Referee comment on "The Met Office operational wave forecasting system: the evolution of the Regional and Global models" by Nieves G. Valiente et al., Geosci. Model Dev. Discuss., https://doi.org/10.5194/gmd-2022-261-RC2, 2023

This paper gives an overview of the wave modelling and forecasting model suite at the Met Office. The paper is well written and describes the various components in some detail.

Our thanks to the reviewer for positive and constructive critiques. Comments have been addressed and we feel these have significantly improved the manuscript.

The manuscript has been modified substantially and special emphasis has been put in: (i) reduction in length, (ii) accuracy of the models with a specific focus on the influence of spatial resolution and wave-current interactions that will lead to further development.

Note that line number corresponds to the revised manuscript with "All Markup".

My main concern is that the wave-current model AMM15 is only very cursorily described. I would want to see a more detailed discussion of why the model seems to have improved directional and wave periods but less accurate significant wave height in regions with strong currents.

A new section on the influence of wave-current interaction on the model accuracy has been added. See Section 5.4 (L386-441).
Additionally, we included a specific paragraph that tried to answer the question of why the model seems to improve direction and wave period but this is not so obvious for significant wave height:
> "Differences in the accuracy of both configurations suggests that wave refraction and shifts in the relative frequency are better captured with the addition of the sea surface currents in most of the domain. However, overall metrics for Hs are slightly weaker in certain areas of analysis such as the Irish and Celtic Seas, English and Bristol Channel and E coast of England. To isolate the effect of currents and not account for any differences in resolution, we run the AMM15SL2 configuration without currents during August 2019 and compare model differences in Hs over two tidal cycles during spring tides (Fig. 10a). Positive residual differences in Hs correspond to those locations where AMM15SL2 presents some degradation respect GS512L4EUK. Model evaluation showed that both configurations tend to slightly overestimate Hs, therefore, the overall positive bias is exacerbated by the contribution of the residual currents in AMM15SL2. Additionally, the evaluation of the currents effects on the wave energy distribution in two different shallow coastal locations demonstrate that including tidal currents produces a consistent shift towards longer periods (Fig. 10e,g) reducing the energy bias between model and observations at low frequencies (not shown), hence the better agreement for the period in AMM15SL2. In terms of Dir, model differences during ebb (Fig. 10b) and flood (Fig. 10c) tide conditions show wave refraction angles of ±10º when currents are included, helping to better capture the distribution of the wave energy in the directional space (e.g., Fig. 8f). This suggests that AMM15SL2 captures the distribution of the energy in terms of frequency and direction better whereas the total energy might be sometimes too large in this configuration. In other words, the bulk energy imparted to the ocean surface waves might be excessive during low-moderate conditions."

More specifically I would want to see a more in-depth discussion of why the authors still think it is worth running the forecast model.

The focus is in the global and regional models that, although not coupled, are the baseline configurations of all the systems currently run operationally at the Met Office. We believe that including a more in-depth evaluation of AMM15 coupled ocean-wave forecasting system with the other models might be too ambitious for a paper (i.e., four models to evaluate), with also further implications such as having to assess a different set of forcing conditions (ECMWF winds). Additionally, the ocean-wave coupled system has been previously described in Tonani et al. (2019) and Bruciaferri et al. (2021), and Valiente et al. (2021b), which demonstrate that model accuracy is almost equal in both coupled ocean-wave AMM15 and wave-only with currents as forcing conditions (similar to an offline coupling).

Regarding why AMM15 wave-only is still a key component in the Met Office forecasting system, we acknowledge that the Met Office wave forecasting products are currently key components for the decision making of multiple organisations and businesses. Many of these clients require systems that provide updated forecast several times a day. This requirement makes the AMM15SL2 UK Waters wave model an essential system that cannot be replaced by the current AMM15 ocean-wave coupled model, a more complex and computationally expensive model only run once a day.

It is mentioned in L524-531 that "For the case of the operational AMM15 ocean-wave coupled with data assimilation, this is currently run once a day providing 5 days forecast. This is still computationally expensive with increased resource demands over the wave-only operational model with currents as forcing that delivers data four times a day. Met Office internal testing demonstrates that a coupled simulation increases 10% the running time per model respect their standalone version; i.e., if an ocean model needs n nodes to run and a wave model needs m nodes, the ocean-wave coupled simulation of the two will need n+m nodes with an increase of 20% in the running time. While studies continue toward a fully coupled prediction system with atmosphere, ocean, land, ice and wave components, the maintenance and development of each of the model components is crucial in NWP."

Minor comments:
Figs 3 and 4, panels g and h, claim to present the standard deviation of the difference between the model and the observations, but I think you are really presenting the standard deviation of the model values alone, NOT the difference. Please confirm.

Corrected. Both Figs read now as:

> L307: "Figure 4. (a,b) Mean, (c,d) bias and (e,f) root mean square deviation (RMSD) between modelled significant wave height (Hs) and merged altimeter observations (MA_SUP03), and (g,h) model standard deviation (SDmodel) across the global domain for GS512L4EUK-AN."

> L312: "Figure 5. (a,b) Mean, (c,d) bias and (e,f) root mean square deviation (RMSD) between wind (U10) forcing conditions and merged altimeter observations (MA_SUP03), and (g,h) model standard deviation (SDmodel) across the global domain for GS512L4EUK-AN."

L 27: You claim that there is a 20% improvement in wave period and direction in nearshore regions. How is this number arrived upon? Compared against which model, AMM15SL2? Please describe the control run for this comparison.

The abstract has been restructured significantly and this statement was removed. A similar statement is now included in Conclusions where the control run is specified. See L546 "…with >20% reduction in the RMSD respect GS512L4EUK."

L 636: tunning -> tuning

Corrected.

L 643: Recent ... Hersbach and Bidlot (2008) is hardly recent.

Reference has been removed and only recent publications were left.

I think the paper can be accepted after these minor revisions.

---

## Author Response (AR2)

I previously understood the purpose of this manuscript to be a description of the Met Office Operational Wave Forecasting System. Therefore, I expect to see a description of the whole system rather than only half of it. I think that a more robust rebuttal of reviewer 1 would have been appropriate. While it is of course true that not everything we write down gets published, so there is a level of uncertainty over what exactly the reviewer is suggesting, the purpose of GMD is to enable peer reviewed publication of such technical details that may be useful to the wider community as well as to future developers of the model. I have not been involved directly in wave modelling since the last century, and I have the attention span of a gnat, but I didn't find the original manuscript too long or difficult.

A possible response to a reviewer finding a manuscript overwhelming is to suggest a rigorous restructuring. Things like a clear description of the structure of the manuscript near the start, updates on the flow between sections at certain points in the manuscript and keeping sections short and focussed can help, on top of removing irrelevant content (or content already published elsewhere that can be cited). I appreciate that your rewriting has already improved some of these aspects. I would like to see the full system described while retaining these kind of improvements.

However, in your response, you agree with the reviewer that removing description of half the system from your description is appropriate. Please can you explain to me how this is OK? Where do you intend to publish the description of the other half of the forecasting system?

We thank the editor for these comments which have helped to significantly improve the manuscript.

Note that line number corresponds to the revised manuscript with "All Markup".

Our original reply to reviewer 1 resulted in a restructuration of the manuscript, including a significant reduction of content. We believe those changes have made the manuscript clearer without compromising reproducibility aspects. We took care to include references to all the models that are part of the Met Office operational system to ensure that the details are easily accessible by the readers. However, we agree with the editor that Section 2, where we describe the system, resulted in a reduced emphasis on AS512L4EUK and AMM15SL2.

Following the editor recommendations, we have now included the description of all Met Office operational configurations (GS512L4EUK, AS512L4EUK and AMM15SL2) as different sections. Please note that AMM15SL2 configuration is used in a wave-only wave model system and in an ocean-wave system. This is introduced in L185 as "AMM15SL2 is the baseline configuration used for the UK waters wave-only and AMM15 ocean-wave coupled models" and both model systems are now described in section 2.3.3 (L185-212). Furthermore, Section 2.4 (L223) also describes the operational production of all 4 models. We maintain the focus of the manuscript on the evaluation on the baseline configurations as the ensemble and coupled systems are already documented in published works. See L303, "For a detailed evaluation of AMM15 Ocean-Wave Coupled Model and AS512L4EUK wave ensemble refer to Saulter (2020b), Bruciaferri et al. (2021), and Bunney and Saulter (2015), respectively.".

The introduction and the abstract have been modified to reflect this change:
- Abstract (L12-17): "The operational system includes a global forecast deterministic model (GS512L4EUK) and two regional models nested one-way covering the Northwest (NW) European shelf and UK waters (AMM15SL2) as well as an Atlantic wave ensemble (AS512L4EUK). GS512L4EUK and AS512L4EUK are based on a multi-resolution four tier SMC 25-12-6-3km grid. The regional AMM15SL2 configuration uses a two tier SMC 3-1.5km grid and is run operationally both as a standalone forced model (includes wave-current interactions) and as the wave component of a two-way ocean-wave coupled operational system."
- Introduction (paper overview; L51): "A description of the operational wave modelling system which includes a global model and two regional models nested one-way covering the Northwest European shelf and United Kingdom (UK) waters and the Atlantic wave ensemble is presented in Sect. 2."

I would also like to see a slightly more technical version of section "2.1 Research to operations" put back in. Practical considerations are critical context. By more technical I suppose that I mean that details such as a meaningful indication of the computer power required to run the system would be very interesting.

A slightly modified version of Section "2.1 Research to operations" (L74) was included again in the manuscript and additional information about the computer power used to run the operational systems has been added. This section now reads as follows:

"All mission critical NWP models at the Met Office are run under an operationally maintained supercomputer production system known as the Operational Suite (OS) which cycles model tasks on a dedicated high-performance supercomputing environment. Since 2016, the Met Office's operational supercomputer has been a Cray XC40 comprising 6212 nodes of Intel Broadwell/Haswell processors with up to 36 cores per node, connected by a high-speed Aries network. As of 2024, this system is due to be replaced by multiple HPE Cray EX systems which together will provide over 3000 nodes of AMD Milan processors with 128 cores per node connected via a high-performance Slingshot interconnect. At the time of writing, the operational GS512L4EUK model runs on 10 Broadwell nodes (360 PEs; processing elements), the AMM15SL2 deterministic on 8 nodes (304 PEs), the AMM15SL2 coupled on 62 nodes (252PEs for the wave component, 1536PEs for the ocean component) and the Atlantic Ensemble on 2 nodes per member (72 PEs).

To maintain consistency and operational resilience, scientific and technical updates to these models follow a prescribed process defined in Parallel Suite (PS) projects, which aim to ensure the successful pull through of scientific improvements of the Met Office's Numerical Weather Prediction Models into the Operational environment. For the upstream NWP modelling systems a PS is essentially a copy of the latest operational suite to which scientific and technical updates are applied. The PS is run in parallel with the current OS for a 6-8 week period during which verification and performance metrics will be collected. The models described here correspond to the latest Met Office operational systems that became operational in May 2022 after PS45, run in parallel with OS44."